# Record summer rains in 2019 led to massive loss of surface and cave ice in SE Europe

Aurel Perșoiu[1,2,3], Nenad Buzjak[4*], Alexandru Onaca[5*], Christos Pennos[6*], Yorgos Sotiriadis[7*], Monica Ionita[8], Stavros Zachariadis[9], Michael Styllas[10], Jure Kosutnik[11], Alexandru Hegyi[5,12], Valerija Butorac[4]

[1]Emil Racoviță Institute of Speleology, Romanian Academy, Cluj-Napoca, 400006, Romania

[2]Stable Isotope Laboratory, Ștefan cel Mare University, Suceava, 720229, Romania

[3]Romanian Institute of Science and Technology, Cluj-Napoca, 400022, Romania

[4]Department of Geography, Faculty of Science, University of Zagreb, Zagreb, 10000, Croatia

[5]Department of Geography, West University, Timișoara, 300223, Romania

[6]Department of Earth Science, University of Bergen, Bergen, 5020, Norway

[7]Department of Geography, Aegean University, Mytilene, 81100, Greece

[8]Alfred Wegener Institute, Helmholtz Center for Polar and Marine Research, Bremerhaven, 27515, Germany

[9]Department of History and Archeology, Aristotle University of Thessaloniki, Thessaloniki, 54636, Greece

[10]École Polytechnique Fédérale de Lausanne, Lausanne, 1015-CH, Switzerland

[11]University of Nova Gorica, Nova Gorica, 5000, Slovenia

[12]Department of History and Archaeology, University of Cyprus, Nicosia, CY-1678, Cyprus

[*]Authors with equal contribution

*Correspondence to*: Aurel Perșoiu (aurel.persoiu@gmail.com, aurel.persoiu@usm.ro)

**Abstract.** Glaciers worldwide are shrinking at an accelerated rate as the climate changes in response to anthropogenic influence. While increasing air temperature is the main factor behind glacier mass and volume loss, variable patterns of precipitation distribution also play a role, though these are not as well understood. Furthermore, while the response of surface

glaciers (from large polar ice sheets to small alpine glaciers) to climatic changes is well documented and continuously monitored, little to nothing is known about how cave glaciers (perennial ice accumulations in rock-hosted caves) react to atmospheric warming. In this context, we present here the response of cave and surface glaciers in SE Europe to the extreme precipitation events occurring between May and July 2019 in SE Europe. Surface glaciers in the northern Balkan Peninsula lost between 17 and 19 % of their total area, while cave glaciers in Croatia, Greece, Romania and Slovenia lost ice at levels higher than any recorded by instrumental observations during the past decades. The melting was likely the result of large amounts of warm water delivered directly to the surface of the glaciers leading to rapid reduction of the area of surface glaciers and the thickness of cave glaciers. As climate models predict that such extreme precipitation events are set to increase in frequency and intensity, the presence of cave glaciers in SE Europe and the paleoclimatic information they host may be lost in the near future. Moreover, the same projected continuous warming and increase in precipitation extremes could pose an additional threat to the Alpine glaciers in southern Europe, resulting in faster than predicted melting.

## 1 Introduction

The recent *IPCC Special Report on the ocean and cryosphere in a changing climate* highlighted the worldwide shrinking of the cryosphere, with ice sheets, mountain glaciers, snow cover and Arctic sea ice all losing mass and volume (IPCC, 2019). Rising temperatures account for most of the recent widespread ice loss (Marzeion et al., 2014), and climate models suggest that warming will continue, leading to unabated glacier melting over the next decades. Small glaciers are most impacted by this recent melting, with climate scenarios suggesting losses of up to 80 % by the end of this century (IPCC, 2019) for glaciers in several regions, including Central Europe. Both mountain and cave glaciers are sensitive indicators of climate change (Oerlemans, 2005; Kern and Persoiu, 2013), but while most studies aim to understand the response of glaciers to past, present and future climate changes, comparatively less attention has been devoted to the role of short-term weather variability and extreme events in ice mass and volume changes (e.g., Hughes, 2008). Particular attention was recently devoted to the role of short-term weather variability, extreme events and climate sensitivity to glacierets and ice patches (Ødegård et al., 2017). Nevertheless, the cited IPCC report does not include information either on glaciers in SE Europe, the region hosting Europe's

southernmost surface perennial ice bodies (Snezhnika and Banski Suhodol glacierets, in the Pirin Mountains, Bulgaria) or on the dynamics of perennial ice accumulations in caves (hereafter "ice caves").

Ice caves occur in mountain regions across the Northern Hemisphere, between 30° N and 77° N, and from sea level to 3300 m above sea level (a.s.l.), in areas where a combination of favorable cave morphology, local topography and climatic conditions allow for the perennial accumulation and preservation of ice (Persoiu and Lauritzen, 2018). With a few exceptions, most ice

caves have a single entrance and descendant morphology, this configuration resulting in a unique type of microclimate, with underground air temperature following external one during winter and being largely independent during summer when external weather-induced temperature changes are not transferred to the cave environment (e.g., Persoiu et al., 2011a). As a direct consequence of the descending cave morphology, negative air temperatures occur throughout the year resulting in continuous accumulation and preservation of ice. These characteristics result in extremely variable winter and very stable summer thermal

conditions. Consequently, ice caves can be described as experiencing weather in winter months and climate during the summer ones. The accumulation of cave ice is the result of either snow diagenesis or freezing of water, or a combination of the two (Persoiu and Pazdur, 2011). Snow diagenesis occurs at the bottom of vertical shafts, in which snow falls and accumulates during winter months. In such caves, ice forms as a combination of snow diagenesis (compaction under its weight) and freezing of meltwater from the surface of the snowpack and from infiltration.

Several studies of European ice caves have demonstrated that the ice can be in excess of 1000 years old (Spötl et al., 2014; Gradziński et al., 2016; Kern et al., 2018a; Kern et al., 2018b), and often preserve important insights on past environmental variability. Over the past decade, various proxies in cave ices have been used to reconstruct temperature variability during the Holocene (Persoiu et al., 2017; Sancho et al., 2018; Badaluta et al., 2020), vegetation dynamics (Feurdean et al., 2011; Leunda et al., 2019) and atmospheric composition (Kern et al., 2009; Kern et al., 2011). Additionally, other studies have targeted the

microbial and fungal communities harbored by cave ice, identifying dormant new species (Brad et al., 2018; Itcus et al., 2019, Paun et al., 2019; Mondini et al., 2019). Unfortunately, research to date has targeted only a handful of ice caves, with the vast majority of existing ones harboring a yet unexplored wealth of information (Persoiu and Lauritzen, 2018).

Whereas ice loss in surface glaciers is mostly due to melting related to rising temperatures (e.g., Marzeion et al., 2014), cave ice ablation is primarily due to drip water delivering heat to the ice (Luetscher et al., 2005; Persoiu et al., 2011a; Colucci et al.,

2016). Therefore, whereas the projected increase in air temperature in mountain areas would result in enhanced mass loss for surface glaciers, the same rising temperatures might only marginally affect ice volume changes in caves. Monitoring studies in ice caves has been done sporadically since the mid-20[th] century (Racovita, 1994; Luetscher et al., 2005; Persoiu and Pazdur, 2011; Kern and Persoiu, 2013; Munroe, 2021), the results showing that reduction of winter precipitation and increase of winter temperatures are the main drivers behind loss of ice, with summer temperatures having a negligible role. Additionally, studies of perennial ice deposits in caves in Norway, Romania and Switzerland have suggested that above average summer precipitation might play an important role in the overall volume changes of cave glaciers (Stoffel et al., 2009; Feurdean et al., 2011, Lauritzen et al., 2018).

Central and southeastern Europe hosts numerous caves with perennial ice. This region has also experienced some of the fastest loss of glacier ice (Zemp et al., 2019, Sommer et al., 2020) over the past decades. Climate models suggest a mixed response in central and southeastern Europe to the global warming trend, albeit with a general decrease in annual precipitation, an increase in winter precipitation and an increase in the frequency and intensity of extreme summer precipitation events (Giorgi et al., 2011; Planton et al., 2012, Giorgi et al., 2016; Topál et al., 2020). All these conditions are important factors in the ablation of ice, both cave ice and glacierets (Kumar, 2011) being particularly sensitive to long-term climatic changes and short-term weather variability (Braithwaite and Raper, 2007; Hughes, 2008; Brown e al., 2010; Colucci and Guglielmin, 2015). Consequently, cave ice deposits are facing the risk of disappearing completely within a decade (Kern and Persoiu, 2013; Kern and Thomas, 2014), leading to the irreparable loss of the diverse records of past climate, environmental conditions and microbial life they record. While accumulation is a slow process (~1-20 cm/year), annual ablation rates could reach values double or even triple that value (Kern and Thomas, 2014, Lauritzen et al., 2018).

In this context, we present the response of surface and cave glaciers in eastern and southeastern Europe to the weather conditions in 2019. We have studied ice caves in Greece, Croatia, Romania, Slovenia and two surface glacierets in Bulgaria and analyzed the ice level and volume changes in response to weather patterns during the spring and early summer of 2019, a period characterized by anomalous atmospheric circulation corroborated with extreme (in terms of amount and intensity) summer rainfall events.

## 2 Glaciers in central and southeastern Europe

Our investigation targeted surface and cave glaciers in central and southeastern Europe (Fig. 1), a region which hosts numerous ice caves in the rugged karst topography of the Carpathian, Dinaric and Rhodope Mountains, including four of the largest underground glaciers in the world (Buzjak et al., 2018; Mihevc, 2018; Pennos et al., 2018; Persoiu and Lauritzen, 2018).

*Scărişoara Ice Cave* (46°29′23.64″ N, 22°48′37.68″ E, Fig. 2a) is located in the Western Carpathian Mountains (Romania), at 1165 m above sea level (a.s.l.). The cave opens to the surface through a ~60 m wide, 47 m deep shaft which continues with a

wide chamber that hosts one of the largest (>100,000 m$^3$) and oldest (>10,500 years) underground glaciers in the world (Persoiu et al., 2017; Perşoiu and Onac, 2019). This ice body primarily formed by freezing of liquid water every early winter, adding a layer of up to 20 cm of newly formed ice to the existing ice body each year. The changes in the volume of the ice are controlled by the amount of water available for freezing in autumn and early winter and by the melting occurring due to rainwater infiltrating inside the cave in summer. Monitoring of ice level changes begun in 1947 (Persoiu and Pazdur, 2011) and showed

a rapid melt of ice during the 1950s due to changes in the morphology of the ice block, followed by alternating periods of ice growth and loss, superimposed on a moderate melting tendency afterwards.

*Chionotrypa Cave* (Mt. Falakro, hereafter *Chionotrypa Falakro*, 41°17′39.9″ N, 24°05′24.2″ E, Fig. 2b) is a 111 m deep alpine cave located at 2080 m a.s.l. in the Rhodope Mountains, northern Greece (Pennos et al., 2018). The entrance of the cave is through a 50 m wide, 65 m deep shaft that enables snow to accumulate at its base in winter. A 3 m high, 15 m wide opening

at the bottom of the eastern wall of the shaft allows access to the lower chambers of the cave. At the bottom of the shaft and continuing through the opening, a 30 m snow and ice deposit has accumulated. The upper 3 m of the deposit consists of compacted snow, while the lower part is solid, layered ice, incorporating rocks and organic matter from the surface.

*Chionotrypa cave* (Mt. Olympus, hereafter *Chionotrypa Olympus*, 40°05′18.7″ N, 22°21′55.5″ E, Fig. 2c) is located on the eastern slopes of Olympus Mountain (Greece), at 2560 m a.s.l. Its entrance is ca. 6.6 by 8 m wide, leading through a vertical

shaft to a snow and ice mass ca. 20 m thick. In winter up to 5 m of snow can accumulate on top of the existing ice block, but it usually melts during spring and summer, the level of ice reaching a minimum in September (Pennos et al., 2018).

*Crna Ledenica* (43°20′52.98″ N, 17°3′5.36″ E, Fig. 2d) cave is located in the Biokovo Mountains (southern Croatia), 6 km from the Adriatic Sea. The cave has four vertical entrances located at elevations between 1477 and 1503 m a.s.l. (Buzjak et

al., 2018): two northern (15x10 m and 7.5x5 m), a middle (5x4 m) and a southern (11x6 m) one. These wide entrances ensure a constant snow accumulation at the bottom of a 60 m long, 25 m wide chamber, as well as cold air circulation during winter (HBSD, 2019, last access: 25 May 2020). The snow, firn and ice deposit occupies an area of about ~450 m$^2$, its thickness ranging from 12 m below the northern and middle entrances to 0.5 m in the more distal parts of the cave. The upper part of the deposit is covered by fresh snow and organic matter collapsing from the surface during winter. Deeper in the cave, the snow is transformed to firn and finally layered ice with embedded rock particles and organic matter. The changes in mass and volume of the ice deposit are controlled by the circulation of cold air between the cave's chamber and the four entrances, snow accumulation and water inflow during warmer, wetter periods (from spring to early autumn).

*Velika ledena jama v Paradani* (Fig. 2e) is located on Trnovski gozd karst plateau in western Slovenia (45°59′19.70 ″N, 13° 50′40.24″ E). The cave is 6534 meters long and 858 meters deep with the main entrance located at 1135 m a.s.l. The entrance opens at the bottom of a doline and leads to a series of interconnected halls, with the first and second containing the perennial ice block. This block has a layered structure, including clear ice alternating with detritus derived from the surface, and it changes from firn to congelation ice with increased distance from the entrance. The maximum depth of the ice block is unknown, but is estimated to an average of 3 meters, suggesting a maximum volume of 8,000 m$^3$ (Mihevc, 2018). The main ice growth periods are in winter (as snow accumulates in the entrance doline) and spring, when snowmelt water freezes to form congelation ice. The main ice loss occurs in summer and autumn following heavy infiltration of rainwater.

In addition to these underground glaciers, we have also studied Europe's two southernmost glacierets, *Snezhnika* and *Banski Suhodol*, located in the Pirin Mountains (Bulgaria) below the northern cliffs of the Vihren (2914 m) and Kutelo (2908 m) peaks (Gachev et al., 2016). Snezhnika glacieret (41°46′09″ N; 23°24′12″ E) is located on an eastward-facing slope and lies at 2440-2490 m a.s.l., whereas Banski Suhodol (41°46′09″ N; 23°23′40″ E) faces north and lies at 2620-2700 m a.s.l. Both glacierets are a legacy of the Little Ice Age (Hughes, 2009) and occupy less than 1 ha (Grunewald and Scheithauer, 2010) with recent geophysical investigations revealing maximum ice thickness of 14 m and 17 m, for Snezhnika and Banski Suhodol, respectively (Onaca et al., 2019).

## 3 Methods

### 3.1 Ice level and volume changes in caves

In 2018, we initiated a research program aimed to preserve the climate memory of vanishing Eastern Mediterranean subterranean glaciers. Ice levels in caves were measured against fixed points on the cave walls and/or estimated using photogrammetry (Table 1).

In Scărișoara Ice Cave, ice level changes were recorded monthly using a dual approach (Fig. 3, Table 1). Firstly, the distance between a fixed point in the rock ceiling directly above the ice ($B_0$ in Fig. 3) and the ice surface was measured using a ruler and secondly, ice level changes at the surface of the ice block were measured on a metal ruler embedded in the ice. The precision was better than 0.3 cm in both cases. The ice level variations at the surface are measured relative to the level of ice in April 1982 ($A_0$ in Fig. 3) when the ruler was placed in the ice (Racoviță, 1994). The results of the first sets of measurements record the distance between the surface of the ice at any given moment ($B_1$ and $B_2$ in Fig. 3) and the rock ceiling, thus being the sum of changes at the surface (climate and weather induced) as well as bottom (induced by basal melting) of the ice block, whereas the later registers only the changes of ice level at the surface of the ice block ($A_1$ and $A_2$ in Fig. 3). Subtracting the latter from the former enabled us to separate changes at the surface which likely resulted from the influence of external meteorological conditions, from changes due to basal melting, the later estimated to be around 1.5 cm/year (Persoiu, 2005). For the purpose of this study, ice level changes induced by external meteorological conditions were considered. Changes in ice volume were continuously monitored in Velika ledena jama v Paradani since September 2009, with the method of wall to ice distance measurements (similar to Scărișoara Ice Cave above) made twice annually. In Chionotrypa Falakro, Chionotrypa Olympos and Crna Ledenica annual ice level fluctuations were intermittently recorded at the end of the ablation period over the past five years. Photographs of the upper surface of the ice and snow body taken intermittently from the same spot at the end of the ablation periods were compared in order to visually estimate the ice level changes. The errors associated with these measurements are less than 0.3 cm in Scărișoara Ice Cave and Velika ledena jama v Paradani, and are estimated to between 20 and 50 cm in Chionotrypa Falakro, Chionotrypa Olympos and Crna Ledenica (Table 1). Thus, the errors for the 2019 ice

level changes are estimated between 1 and 25 % (Table 1), lower where rulers were used and higher where the presence of snow precluded precise measurements and/or estimates.

## 3.2 Glacieret area and volume changes

To analyze in detail the surface changes of Snezhnika and Banski Suhodol glacierets, two UAV (DJI Phantom 4 Pro drone equipped with 20 megapixel camera) surveys were performed at the end of the ablation season (October 2018 and September

2019). The drone survey was designed to cover the glacierets and their surroundings. For flight planning and mission control we used DJI Ground Station Pro software. The flight height was set at 200 m and the images were collected every three seconds along parallel lines with an overlap of 80 %. Ground control points (GCPs) were used for both glacierets (five for Snezhnika glacier and eight for Banski Suhodol glacier) to georeference the digital surface models and orthophotos. The GCPs were measured with a high accuracy Hiper V Topcon real-time kinematic (RTK) positioning system. The images were processed

using Agisoft Photoscan Professional with the following workflow: 1. Alignment and match of photos, 2. Analysis of the sparse cloud, 3. Import and setting of the GCPs on each camera and the free-network bundle adjustment, 4. Generation of the referenced dense cloud with medium accuracy settings and moderate depth filter, 5. Generation of mesh and textures, 6. Creation of the Digital Surface Model, 7. Creation of an orthophotomosaic. Using this workflow, high-resolution digital surface models were created for both glacierets (Table 2).

## 3.3 Weather and climate


As most of the investigated caves are located far from meteorological stations, or existing station data covers inappropriate time periods for this study, we extracted meteorological data from the E-OBS dataset (Cornes et al., 2018). The E-OBS data set builds on the European Climate Assessment and Dataset (ECA&D), which is a database of daily meteorological station observations across Europe. The station based data used to develop the gridded E-OBS data set are subjected to quality control

and homogeneity tests, thus, all series are checked for inhomogeneities. Nevertheless, the database is still prone to uncertainties related to the measurements and the spatial interpolation. Compared to previous versions of the E-OBS dataset, the one used in this study is based on 100 realizations of each daily field (e.g., precipitation, daily mean temperature, daily maximum

temperature and daily minimum temperature). The spread of the E-OBS dataset is calculated as the difference between the 5th and 95th percentiles over the 100 realizations to provide a measure indicate of the 90% uncertainty range. For the current study we use the ensemble mean (the mean over the 100 realizations), which provides a "best guess" value. To look at the intensity and magnitude of the summer precipitation we use the highest 5-day precipitation amount (RX5day), which was computed based on the EOBS-v20e data set (Cornes et al., 2018), with a spatial resolution of 0.1 x 0.1. The ranking maps of 2019 were made for RX5day using the six first ranks. The rank maps are obtained by comparing the magnitude of the variable of each month (May–July 2019 for the present study) relative to the same month in each year from 1950. A rank of one implies record-breaking precipitation in 2019; a rank of two indicates that 2019 had the second most extreme value in that month (since 1950) etc.

In order to understand the role of large scale circulation patterns in determining specific weather types, we have computed the anomalies of the mean air temperature (TT) and geopotential height at 500 mb (Z500) and we analyzed the conditions of snow cover and extreme precipitation (monthly maximum consecutive 5-day precipitation, RX5day) for both the ice accumulation (December 2018–February 2019) and ablation (May–July 2019) periods. The monthly mean temperature was extracted from the E-OBS data set (Cornes et al, 2018). The Z500 file was extracted from the NCEP/NCAR 40-year reanalysis project (Kalnay et al., 1996). The snow cover data were provided by MODIS/Terra Snow Cover Monthly L3 Global 0.05Deg CMG, Version 6 (Hall et al., 2006).

## 4 Results

### 4.1 Ice level, volume and area changes

Ice level was constant in *Scărişoara Ice Cave* between July 2018 and September 2018 when a shallow lake started to form on top of the ice block. The water of this lake froze in early December 2018, adding a layer of 15 cm to the existing ice. Heavy snowfall in the 2018-2019 winter and subsequent melt in spring 2019 led to rapid infiltration of large volumes of water inside the cave, resulting in ca. 5 cm of ice being formed in the still undercooled cave environment. Between May and July 2019, the ice level in Scărișoara Ice Cave dropped by ca. 35 (±0.3) cm (Fig. 4), due to continuous infiltration of surficial warm water.

This catastrophic melt event spread across the entire upper surface of the 3000 m$^2$ ice block, resulting in the loss of ca. 1050 ($\pm$90) m$^3$ of ice.

In *Chionotrypa Falakro*, a gradual decrease of the ice volume was evident since 2014, reaching a minimum in September 2019 (Fig. 5). The ice level at the bottom of the entrance shaft fell ~3 m and receded ~1 m from the cave walls between July 2018

and September 2019. A rough estimation suggests that >600 ($\pm$60) m$^3$ of cave ice were lost during this period.

In *Chionotrypa Olympus*, the volume change of the snow and ice deposit was strongly positive in winter 2018/2019, but steadily decreased in summer 2019. Compared to the situation in 2018, ice and snow were lost both at the surface of the deposit and in the rimayes surrounding it, amounting to a loss of ca. 15–30 ($\pm$3.75–7.5) m$^3$ of the total ice volume.

In *Crna Ledenica* observations in 2019 show that following the cold and wet 2018–2019 winter large amounts of fresh snow

accumulated below the entrance shafts. Further, meltwater infiltration in spring originating in thick winter snowpack led to the development of numerous ice speleothems and ice crusts. However, heavy summer rains and subsequent infiltration of warm water resulted in the complete melting of these ice crusts by September 2019. The ice-covered area and the total volume of perennial ice also retreated in summer 2019. In some parts of the cave, the ice thickness dropped by up to ~2 m, and the total area covered by perennial ice and snow dropped by ca. 200 m$^2$, resulting in a loss of about 150-200 ($\pm$15-20) m$^3$ during summer

225    2019.

The deeper parts of *Velika ledena jama v Paradani* cave were less affected by ice volume loss, compared with the entrance area. The ice slope connecting the inner part of the cave with the entrance has retreated by several meters towards the cave's entrance and an almost continuous surface of broken host rock developed on the former ice surface. Since June 2013 the ice surface lowered by 220 ($\pm$2) centimeters, of which 41 ($\pm$0.3) centimeters were lost between June 2018 and June 2019.

In October 2018, Snezhnika glacieret extended over 5700 m$^2$, compared with 4600 m$^2$ in September 2019 (Figs. 6a and 6b), while Banski Suhodol glacieret extended over 11600 m$^2$ in October 2018 (Fig. 6c) and 9600 m$^2$ in September 2019 (Fig. 6d). Between 1959 and 2008 Snezhnika glacieret lost almost half of its surface (from 13000 to 7000 m$^2$, Grunewald and Scheithauer, 2010), and between 2008 and 2018 lost another 1300 m$^2$ while during 2019, it lost 1100 m$^2$. The average annual loss of Snezhnika glacieret between 1959 and 2018 was rather constant between 122 and 130 m$^2$ year$^{-1}$, but dramatically

increased by one order one of magnitude (1100 m$^2$ year$^{-1}$) in 2018-2019. The loss of ice at the surface of these glaciers

(excluding basal melting) between 2018 and 2019 amounts to 10045 (±241) m$^3$ (Banski Suhodol) and 2933 (129) m$^3$ (Snezhnika). Aerial footage indicates a general retreat at the lower end of the two glaciers and the increase in the width of the rimaye separating them from the cirque headwall (Fig. 6). Additionally, in 2019 a small glaciated area became separated from the main ice body of the Banski Suhodol glacieret, and two more indentations are likely to cause rapid disintegration of the main ice body in the near future (Fig. 6d). Between 2018 and 2019, the termini of both glacierets show significant loss of ice. The mean retreat of the terminus was 3.4 m at Snezhnika and 5.8 m at Banski Suhodol (Fig. 7). Fig. 7 (c, d) shows the difference in the elevation of the ice surface and the surrounding area between 2018 and 2019. In general, the difference in Snezhnika ice surface elevation between 2018 and 2019 ranges between 0.2 and 1 m, exceeding 1.2 m only in the southwestern part. On the other hand, more than half of Banski Suhodol surface has lowered by more than 1 m due to surface melting. Downslope of the glacierets, several snow patches present in 2018 disappeared in 2019.

## 4.2 Meteorological data

Meteorological data (Fig. 8) show that 2019 was exceptionally wet in SE Europe. At all investigated sites, precipitation amounts exceeded the long-term (1971–2000) average by more than 150 %. Following a relatively dry early spring (compared to the 1971–2000 average), precipitation amount started to increase in April 2019, peaking in May, June and July (Fig. 8). In May 2019, all the analyzed cave ice locations received more than 100 % of the climatological mean precipitation (1971-2000). Crna Ledenica and Chionotrypa Falakro received much more, at 250 % and 200 %, respectively.

In December 2018, the geopotential height anomalies at 500 mb level were characterized by a wave-train like structure with negative Z500 anomalies (indicative of a low-pressure system) over the central North Atlantic Basin, positive Z500 anomalies (indicative of a high-pressure system) over the central part of Europe extending northwards, and negative Z500 anomalies over the eastern part of Europe (Fig. 9a). The low-pressure system over eastern Europe, which was carrying moist air from the Black Sea together with cold and dry air from the north, led to snowfall events over the high altitudes in the Carpathian Mountains and some parts in Ukraine (Fig. 9b). Overall, December 2018 was warmer than normal over the whole European region with some exceptions in the Alpine areas and Bulgaria (Fig. 9c).

In January 2019, the large-scale atmospheric circulation was characterized by a dipole-like structure with positive Z500 anomalies centered over the central North Atlantic basin and negative Z500 anomalies centered over Europe (Fig. 9b). This dipole-like structure led to several episodes of snowfall in the Alps and the Carpathian Mountains, due to the advection of northerly cold air from the Arctic region coupled with moist and warm air intrusions from the Atlantic basin. In January 2019 most of central and eastern Europe were snow covered (Fig. 9e) and the Alpine region was up to 5 °C colder than normal (Fig. 9f).

In February 2019, the prevailing large-scale atmospheric circulation was reversed compared to the previous month. A low-pressure system prevailed over the central part of the North Atlantic basin, while Europe was under the influence of a high-pressure system (Fig. 9g), resulting in dry and warm weather (Fig. 9i). Snow was present only over the mountain regions in the Alps and the Carpathian Mountains (Fig. 9h). Overall, February 2019 was exceptionally warm in the northern and eastern part of Europe (European State of the Climate, 2019).

In May 2019, the central and southeastern parts of Europe were characterized by below average air temperatures. The prevailing large-scale atmospheric circulation featured a Rossby wave guide with negative Z500 anomalies over the eastern U.S. coast, positive Z500 anomalies over the central North Atlantic basin and western part of Europe, negative Z500 anomalies over the eastern part of Europe and positive Z500 anomalies over Russia (Fig. 10a). Concurrently, Italy, the Alpine regions, Croatia and the northwestern part of Romania experienced extreme rainfall (Fig. 10b). Rainfall records were broken over small areas in the Alps and Ukraine (Fig. 10c). Most of the intense and record-breaking rainfall was recorded over the regions were the investigated glaciers are located (Fig. 1).

In June 2019 most of the European continent was under the influence of a high-pressure system (Fig. 10d) that led to the advection of warm and dry air from Africa and the subsequent development of a record-breaking heat wave over the southeastern and central parts of Europe (European State of the Climate, 2019). June 2019 was dry over large areas of Europe, with some small exceptions over the Alpine region and the southeastern region (Fig. 10e), but with record rainfalls in Bulgaria and Greece limited to relatively small areas (Fig. 10f). The precipitation in the southern part of Europe mainly occurred during heavy thunderstorms, resulting in single day rainfall events of ~50 mm locally.

As in June 2019, in July 2019 most of Western Europe was under the influence of a high-pressure system (Fig. 10g) and was very dry over large parts of Europe (particularly the western half). Wetter conditions, with enhanced rainfall, were recorded in southeastern Europe (including Croatia and Slovenia). The rainfall over these regions was mainly due to heavy thunderstorms (Fig. 10h).

## 5 Discussion

All ice caves in the investigated region are located well below the 0 °C MAAT (Mean Annual Air Temperature) isotherm. Contrary to high-altitude glaciers, underground glaciers have both the accumulation and ablation zones in the same location, in most cases at the bottom of vertical shafts where snow and ice accumulates in winter but also melts in summer (e.g. Perșoiu and Lauritzen, 2018), thus making them extremely sensitive to both short and long term climatic changes, and especially to heat delivered by warm rainwater.

The 2018-2019 winter was extremely favorable for ice accumulation in caves in SE Europe, following widespread snowfall (Fig. 9e) that likely resulted in thick snow cover on the ground (Fig. 11) and consequently contributed to volume gains of cave and surface glaciers. In January 2019, heavy snowfalls in the eastern half of Europe (e.g., Fig. 9e) resulted in large accumulation of snow in Chionotrypa Falakro, Chionotrypa Olympus and Crna Ledenica caves (as observed in the field) and likely on the surface glaciers in Bulgaria. In Romania and Slovenia, higher than average temperatures in February 2019 (Fig. 9i) led to the rapid melt of the surface snowpack resulting in melt water infiltration in Scărișoara Ice Cave and Velika ledena jama v Paradani, where thermal conditions were still negative, resulting in rapid formation of ice layers (Perșoiu et al., 2011b). Ice level measurements indicate that a 15 cm thick layer of ice was added to the upper surface of the ice block in Scărișoara Ice Cave, exceeding the mean annual growth for the 2000-2018 period (Perșoiu and Pazdur, 2011; Perșoiu, 2018). Overall, in the 2018-2019 winter northerly cold air advection and westerly and southerly moisture transport (Fig. 9) resulted in rapid ice accretion in caves in SE Europe, both as direct snow accumulation and water freezing.

However, winter ice accumulation was counterbalanced in 2019 spring and summer, when higher than average precipitation led to rapid cave ice ablation in Crna Ledenica and Chionotrypa Falakro caves as well as the glacierets in Bulgaria. At all these locations precipitation amounts between May and July 2019 exceeded the multiannual mean (Fig. 10), mainly during extreme

thunderstorm events (Figs. 10b, e, h). These extreme events delivered large volumes of warm water directly to the surface of cave and surface glaciers, leading to rapid melting. In the case of Scărișoara Ice Cave and Velika ledena jama v Paradani, where the main ice blocks are not located directly below the caves' entrance shafts, similarly extreme summer thunderstorms (Figs. 8 and 10) resulted in high volumes of water entering the caves through fissures in the host rock as percolating water. This led to widespread melting of ice, resulting in the loss of about 35 cm of ice over the entire surface (>2000 $m^2$) of the ice block in Scărișoara Ice Cave (Fig. 4). On Mt. Olympus, summer 2019 was warm and dry (Fig. 10). In Chionotrypa Olympus, the surface of the ice is just 6 m below the cave entrance (Pennos et al., 2018) and thus the cave ice deposit responds to climatic variability in a manner similar to surface glaciers. The thermal inversion layer inside this shallow entrance shaft was destroyed during the prolonged warm spell (as observed during visits to the cave), triggering the rapid melt of the surface and sides of the glacier; however, lack of precise measurements prevents us from estimating a figure for the volume of lost ice.

Cave ice deposits are prone to rapid disintegration once melt-water channels start to develop on their surface, channels that 1) drain away water before it would freeze to form new layers of ice (Persoiu and Pazdur, 2011) and 2) enhance melting and fragmentation leading to rapid loss of ice (e.g., Lauritzen et al., 2018). The cumulative ice loss in summer 2019 for the four analyzed caves is ~1300 $m^{-3}$. This loss adds to the long-term worldwide melting of cave ice (Kern and Persoiu, 2013), but it is unprecedented over the past century (Fig. 12). Several studies (Kern et al., 2009; Persoiu and Pazdur, 2011) of long-term links between ice volume changes and climate in caves have shown that cave glaciers have a non-linear response to air temperature changes. These studies suggest extreme warming events play an insignificant role in the melting of ice, with warm water infiltration and the length of the warm season instead being the most important factors causing melting. Similar observations were made by Colucci et al. (2016) in the southern Alps, where rapid and massive loss of cave ice was documented as a result of melting under the influence of warm water advection delivered by extreme (in terms of amount and intensity) precipitation events.

A behavior similar to the ice caves in Greece and Croatia where the ice and snow deposits are directly influenced by external weather and climatic conditions was observed for the Snezhnika and Banski Suhodol glacierets. The rapid melting of these glacierets was accelerated by the warm and moist conditions resulting from turbulent heat fluxes near the surface of the ice (Marks et al., 1998; Pomeroy et al., 2016), in addition to the heat delivered directly by rainwater. The relative area loss of

Snezhnika glacieret was ~ 0.94 % $a^{-1}$ between 1959 and 2008, similar to the average global value of 1 % $a^{-1}$ (Vaughan et al., 2012), but increased to 1.86 % $a^{-1}$ between 2008 and 2018 and 19.3 % $a^{-1}$ in 2019. For Banski Suhodol glacieret, the relative area loss between 2018 and 2019 was close to that of Snezhnika glacieret at 17.2 % $a^{-1}$. The dramatic increase in area loss, coupled with the overall shallow thickness of these glaciers makes them especially vulnerable to rapid disintegration. The temperature/precipitation equilibrium line altitude (TP-LEA) for surface glaciers and glacierets in southeast Europe is well above the highest peaks (Hughes, 2018) so they are in a state of continuous loss of ice, with extreme events like those described here threatening their survival. Extrapolating our data, episodes of rapid summer melt induced by similar extreme events, either heat waves (Hughes, 2008) or precipitation (this study) may result in the loss of surface ice by 2035 CE. Similarly to the studied Bulgarian glacierets, perennial snow patches on Mt. Olympus, remnants of glaciers from the last glacial cycle (Styllas et al., 2018), began to disintegrate in 2019 under the prolonged heat wave. These cases mirror recent findings from SW Europe, where Moreno et al. (2020) have shown that glaciers that survived warm periods during the past 2000 years are rapidly melting. All surface glaciers in southern Europe are out of balance with present-day climatic conditions, but the slow melting occurring at their termini generally results in gradual re-equilibration (Zekollari et al., 2020). However, recent rapid warming leads to an increase in the altitude of the 0 °C isotherm (Rottler et al., 2019) thus further increasing the imbalance between glaciers and climate and enhanced melting. Our results suggest that, adding to the melting under increased temperatures, heavy summer precipitation events result in enhanced melting of both cave and surface glaciers. With increasing temperatures, the altitudinal rise of the 0 °C isotherm (Rubel et al., 2017) would bring more glaciated terrain under warming conditions, and thus yet more susceptible to heat transfer during heavy summer thunderstorms and extreme summer heat waves. Accelerated warming of the Arctic (Cohen et al., 2020) would result in meridional amplification and slower propagation of the Rossby waves, leading to an increase in the frequency of blocking conditions and associated extreme events (Francis and Vavrus, 2012; Liu et al., 2012; Screen and Simonds, 2014). The increased frequency, duration and intensity of both heat waves (e.g., Spinoni et al., 2015) and heavy rainfall events (Púčik et al., 2017; Rädler et al., 2019) in southeast Europe would thus lead to a higher ablation rate of surface and cave glaciers than that expected from increased temperatures alone. Especially vulnerable are cave glaciers, already located in areas subject to both warming and extreme summer thunderstorms, and surface glaciers close to the 0 ° isotherm, thus resulting in the loss of ice faster than predicted by the most recent estimates (IPCC, 2019; Paul et al., 2020). The massive

loss of ice surface for the Bulgarian glacierets in one single year (17-19 % a$^{-1}$) coupled with the increased occurrence of the most intense precipitation events (Myhre et al., 2019) and the positive feedback processes following disintegration of small glaciers (e.g., Paul et al., 2004) suggest that this could happen before the middle of the next decade.

**6 Conclusions and outlook**

We have investigated the response of cave and surface glaciers to extreme summer rain events in SE Europe during 2019 and unraveled unprecedented ice loss over the observational period. Surface glaciers in the northern Balkans lost nearly one fifth of their surface in a single summer, while caves lost ice that accumulated during several decades. At this rate, mountains in SE Europe could be glacier free by 2035 CE. The ice loses are related to enhanced melting resulting from outsized amounts of

water reaching the caves and delivering large amounts of heat directly to the ice. The synoptic conditions leading to these extreme events were induced by the persistence of blocking conditions over Western Europe leading to extreme heat waves in southern Europe and record-breaking amounts of rainfall. While cave and surface glaciers in mountains across Europe are sensitive to increasing temperature, we have shown that extreme summer rain events may be more important. These events lead to rapid melting and disintegration of ice bodies, rendering them even more sensitive to temperature changes. As climate

models suggest future changes in the dynamics of Rossby waves leading to more extreme events, disappearance of both surface and cave glaciers in SE Europe (and elsewhere) may occur earlier than predicted.

The ongoing and predicted loss of recently accumulated ice threatens the possibility to accurately reconstruct past climate variability using various proxies harbored in cave ice. In order to generate (semi) quantitative reconstructions it is crucial to collect ice that grew during the instrumental period (the last ~50-100 years) and compare the reconstructed variables with

measured ones. The accelerated melting of ice is quickly reducing the possibility to perform such studies, thus leading to the additional loss of invaluable climatic information.

*Data availability*. The E-OBS data set was downloaded from the European Union's Earth observation programme available at https://surfobs.climate.copernicus.eu/dataaccess/access_eobs.php#datafiles (last access on March 23, 2021). The snow depth

data was provided by Deutscher Wetterdienst (www.dwd.de, last access on March 23, 2021).

*Author contributions.* AP designed the study and wrote the manuscript with input from all authors. MI analyzed the climate data. CP, YS and SZ collected data and YS reconstructed the dynamics of ice in Chionotrypa Cave (Falakro), NB and VB made measurements in Crna Ledenica Cave, MS collected data from Chionotrypa Cave (Olympus), JK measured ice dynamics

in Velika ledena jama v Paradani and AO and AH collected and analyzed data from the surface glaciers in Bulgaria.

*Competing interests.* The authors declare that they have no conflict of interest.

*Acknowledgments.* The work behind this article would not have been possible without fieldwork support from Nina Marić,

Željko Marunčić Bospor, Ioanna Mylona, Tasos Polihroniadis and Matea Talaja. We thank Toni Tursić-Franky for hosting us during fieldwork in Croatia and the Administration of the Apuseni National Park (Romania) and the Public Institution Biokovo Nature Park (Croatia) for the continuing support. The Ephorate of Palaeoanthropology and Speleology of the Greek Ministry of Culture is thanked for providing access to Chionotrypa Cave. Fieldwork in Greece and Croatia was partly supported by the International Union of Speleology. AP was financially supported by UEFISCDI Romania through grants no. PN-III-P1-1.1-

TE-2016-2210 and PN-III-P4-ID-PCE-2020-2723. MI is supported by Helmholtz funding through the joint program "Changing Earth - Sustaining our Future" (PoF IV) of the AWI. Funding by the AWI Strategy Fund Project - PalEX and by the Helmholtz Climate Initiative - REKLIM is gratefully acknowledged. AO received financial support from the Ministry of Research and Innovation, CNCS - UEFISCDI, project number PN-III-P1-1.1-PD-2016-0172, within PNCDI III and AH was financially supported by the Romanian grant PNIII28 PFE BID. The research leading to these results has received funding

from the EEA Grants 2014-2021, under Project contracts no. 3/2019 (KARSTHIVES) and 4/2019 (GROUNDWATERISK). IP-SP project SPIRIT251220 provided support during the revision of the manuscript. We thank Dr. Sevasti Modestou who assisted with editing of our manuscript. Ketil Isaksen, Zoltán Kern and two anonymous reviewers provided useful comments that helped us reshape the manuscript.

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

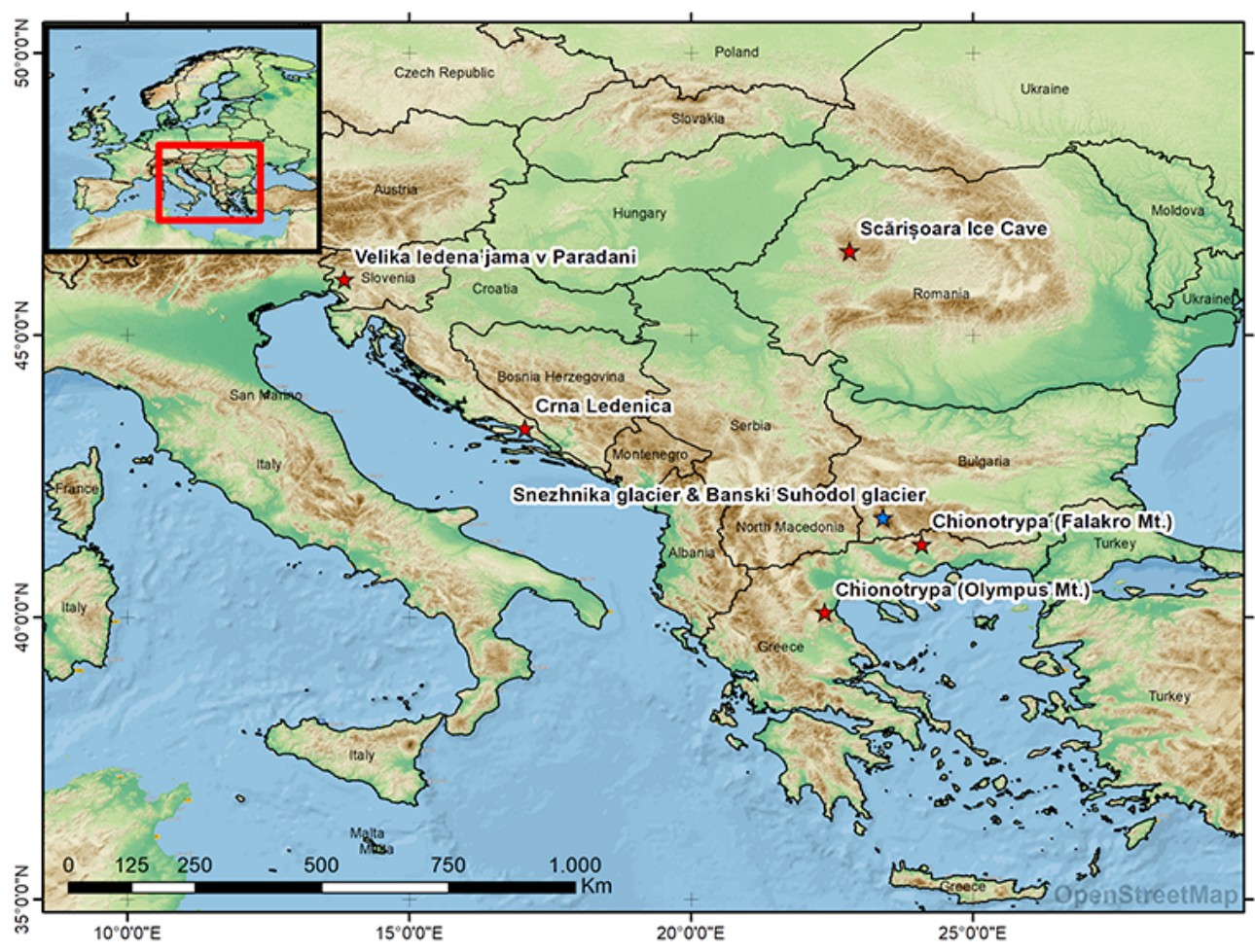

**Figure 1. Location of investigated ice caves (red stars) and surface glacierets (blue star). Base map © OpenStreetMap contributors 2020 distributed under a Creative Commons BY-SA License.**

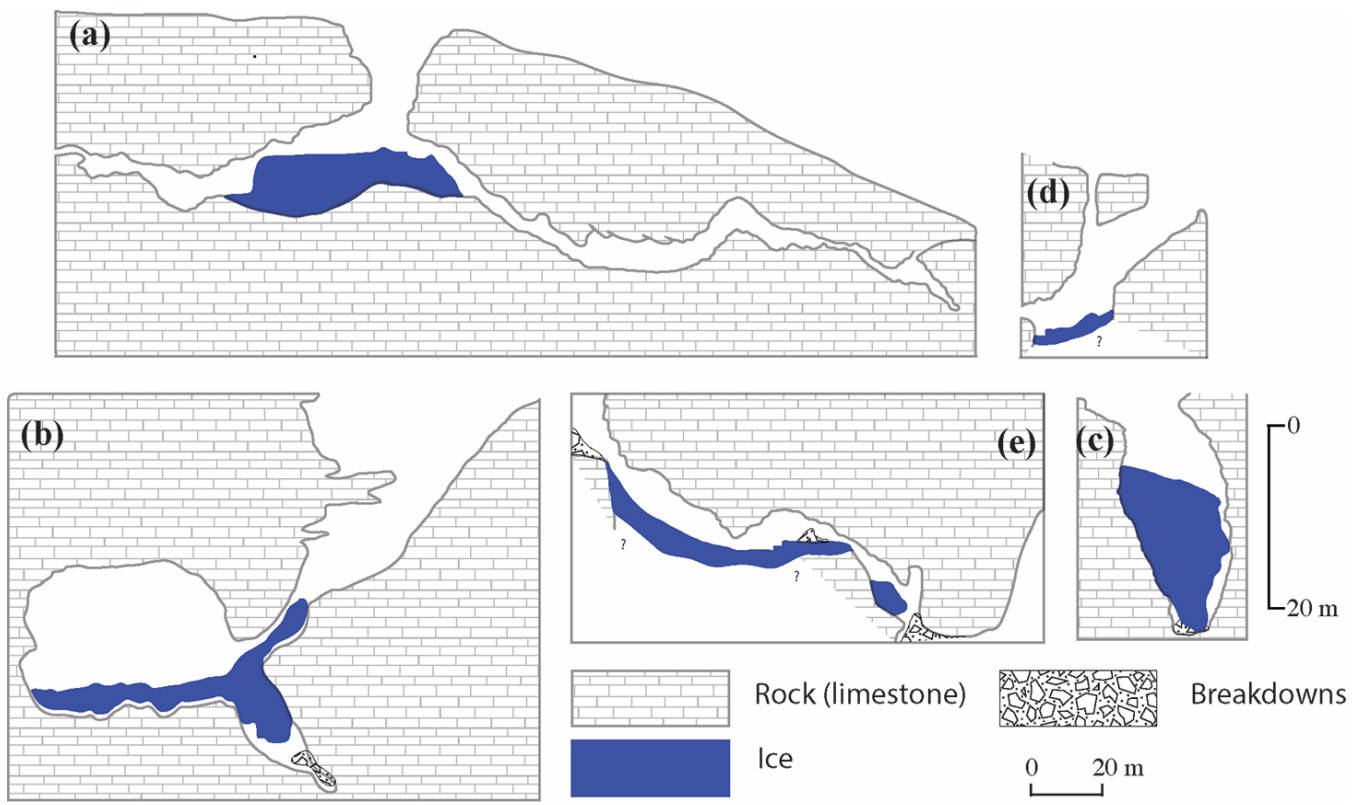

Figure 2. Cross-sections through the investigated caves: a) Scărişoara Ice Cave (modified from Rusu et al., 1970), b) Crna Ledenica (map by N. Buzjak), c) Chionotrypa Olympos (map by M. Vaxevanopoulos, modified from Pennos et al., 2018), d) Chionotrypa Falakro (map by Y. Sotiriadis, modified from Pennos et al., 2018), e) Velika ledena jama v Paradani (modified from a map by Kunaver Pavel). All maps share a similar scale (on the bottom of the panel), except Chionotrypa Olympos, for which a separate vertical scale (half the size of the previous one) on the right of the figure applies.

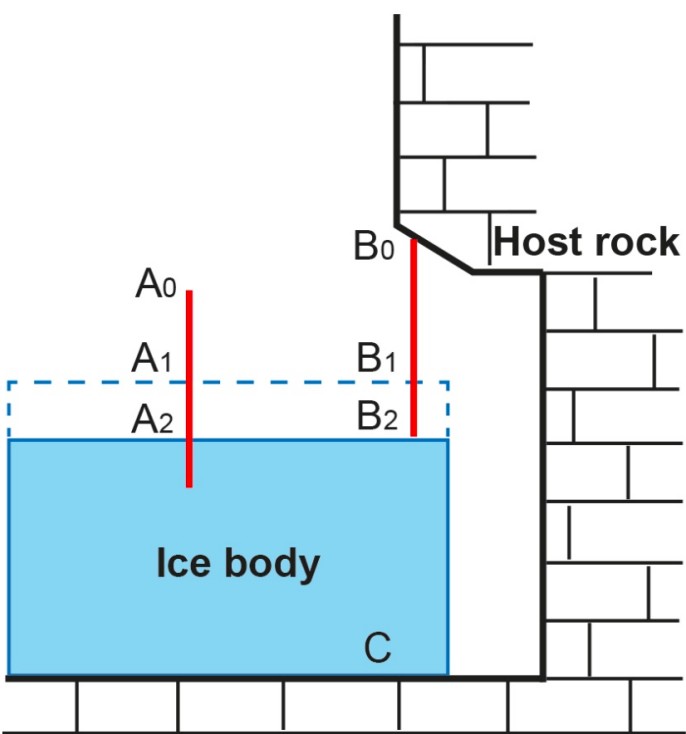

**Figure 3. Ice level change measurements in ice caves. The scale A is fixed in ice and moves together with it, as basal melting (C) affects the base of the ice body. Lengths $A_0$-$A_1$ and $A_0$-$A_2$ quantify changes in ice level at the surface only. B0 is a fixed point in the rock ceiling above the ice. Lengths $B_0$-$B_1$ and $B_0$-$B_2$ record ice level changes at the surface of the ice block *plus* changes at the bottom (C).**

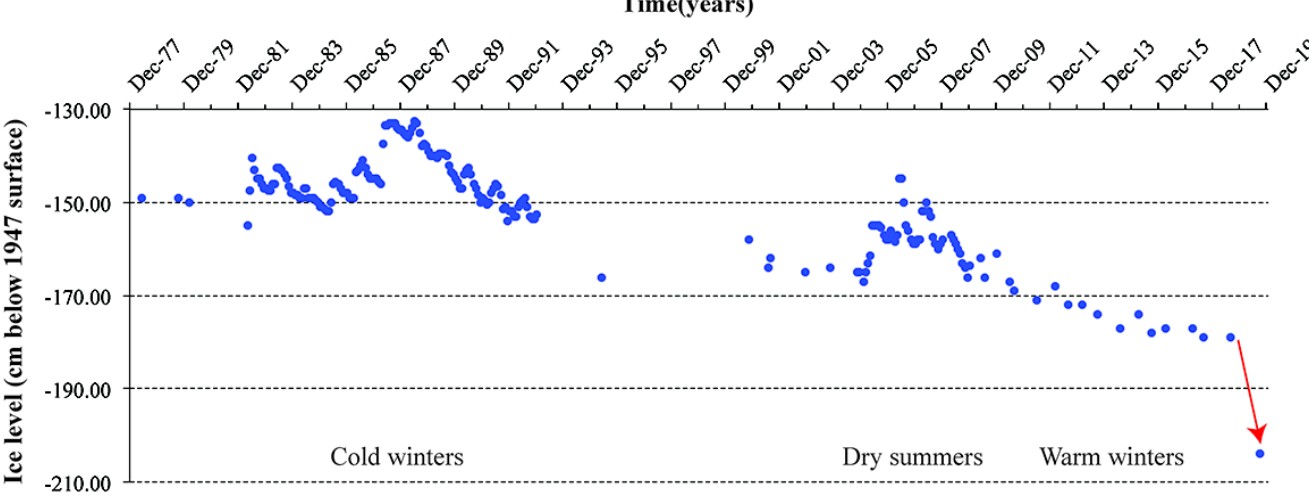

**Figure 4: Upper surface ice level changes of the glacier in Scărișoara Ice cave, Romania, between 1975 and 2019**
(**updated from Perșoiu and Pazdur, 2011). The red arrow points to the unprecedented annual melt in 2019.**

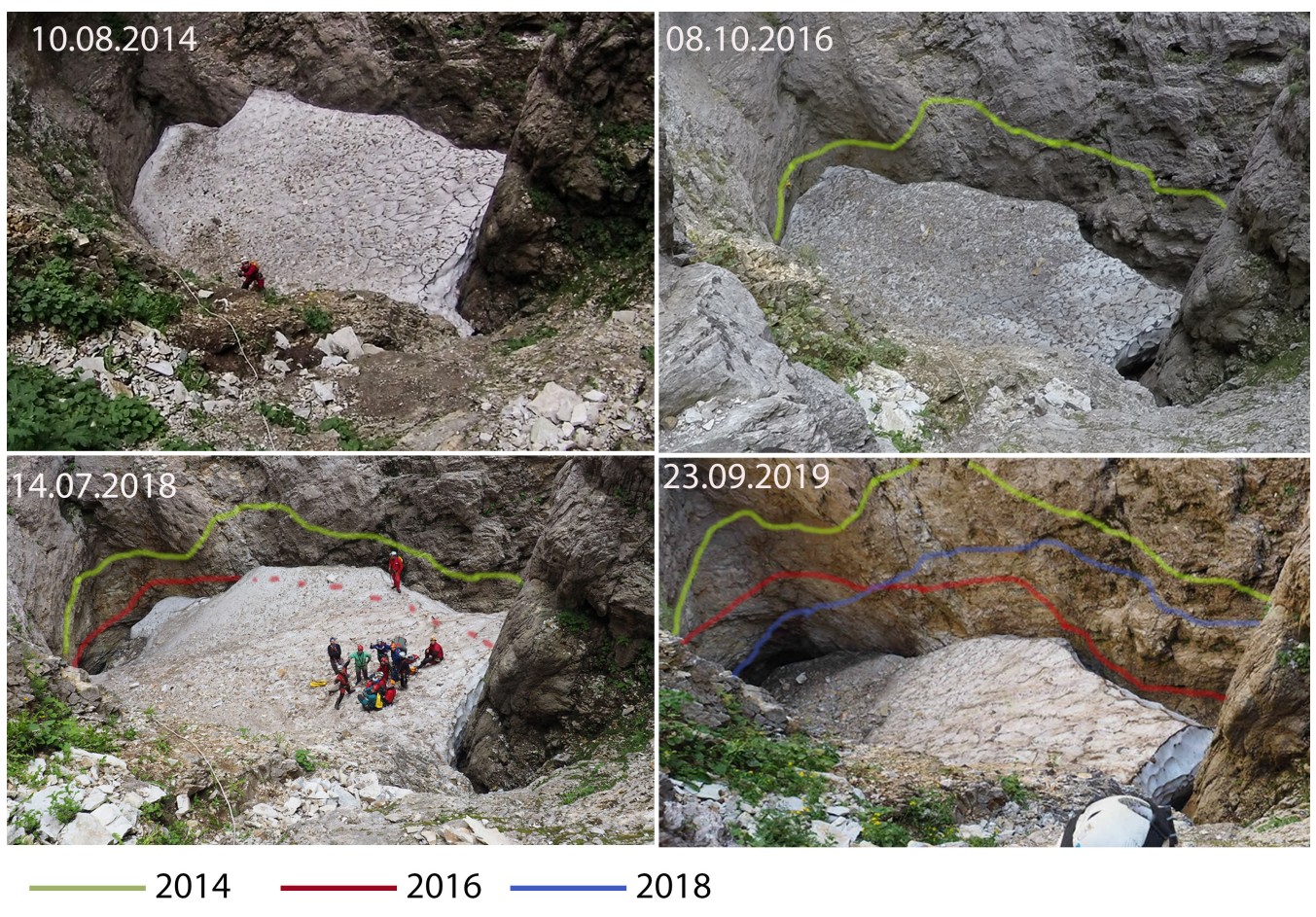

**Figure 5. Changes in the surface morphology of snow and ice in Chionotrypa Cave (Falakro Mountain, Greece), 2014 - 2019. Photo credit: Y. Sotiriadis.**

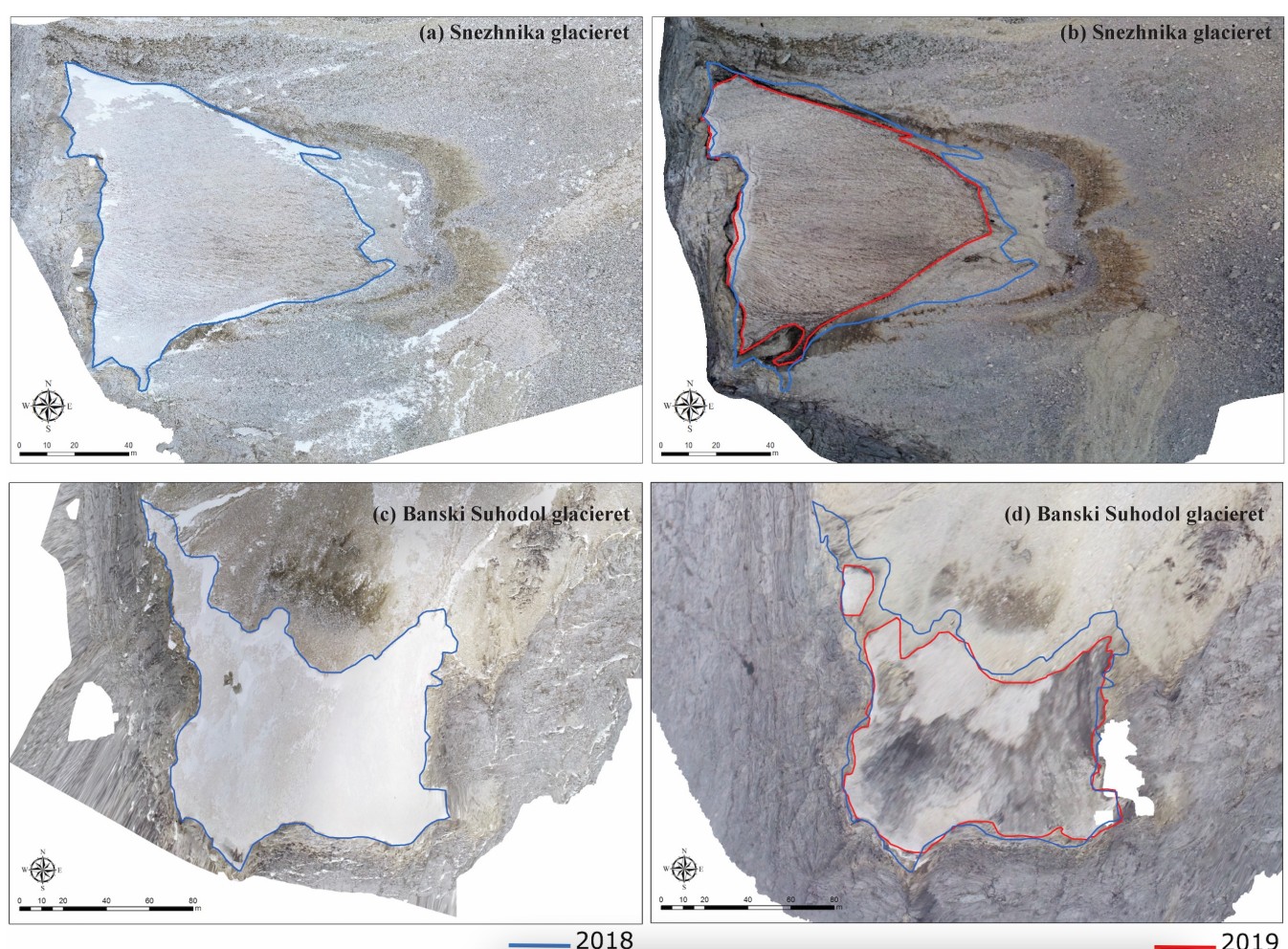

**Figure 6. Orthomosaics showing ice surface changes for Snezhnika (a and b) and Banski Suhodol glacierets (c and d), Bulgaria, between 2018 (a and c) and 2019 (b and d).**

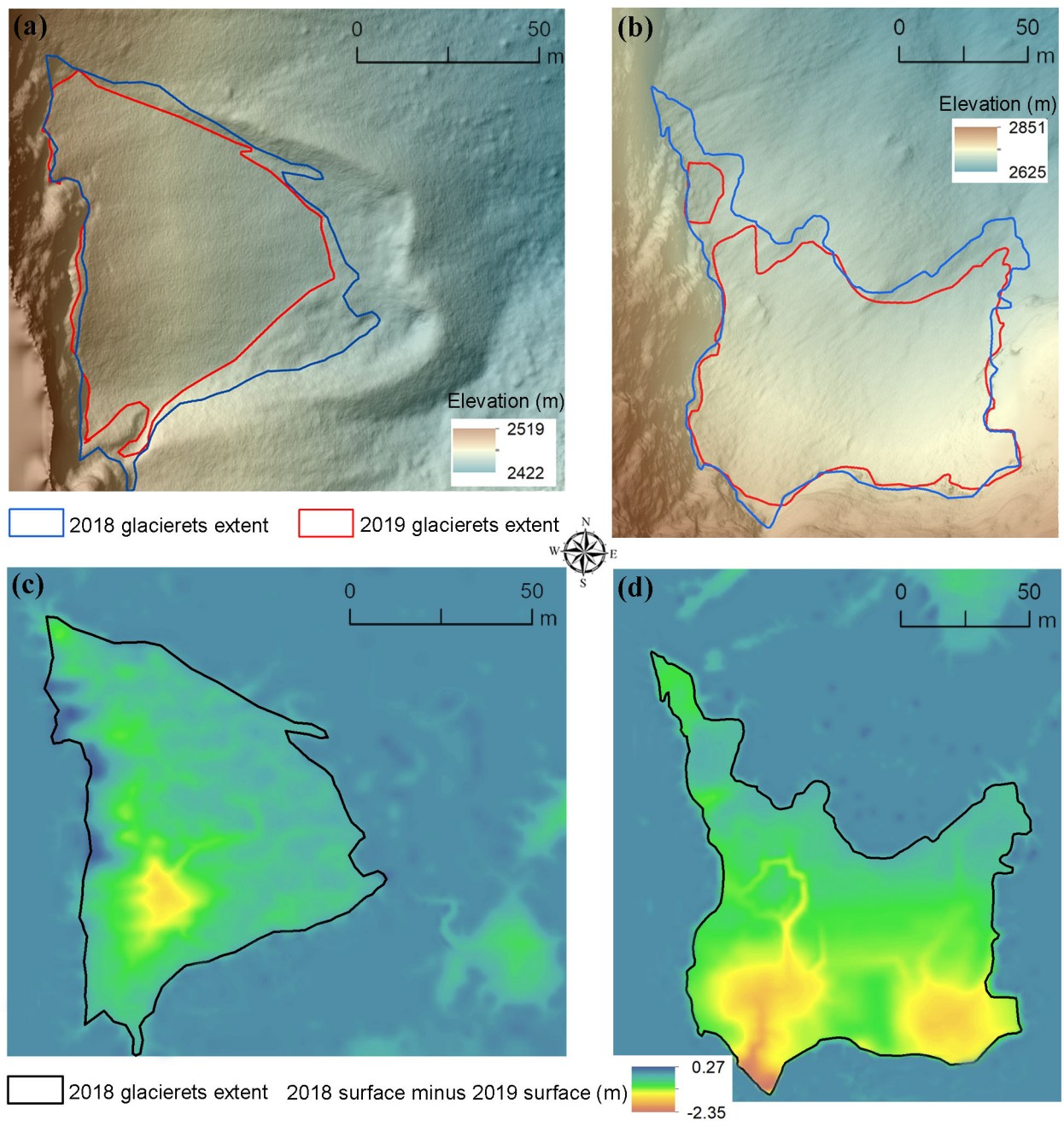

**Figure 7. Digital elevation models of Snezhnika (a) and Banski Suhodol (b) glacierets in 2019 and changes in ice surface elevation between 2018 and 2019 at Snezhnika (c) and Banski Suhodol (d) glacierets.**

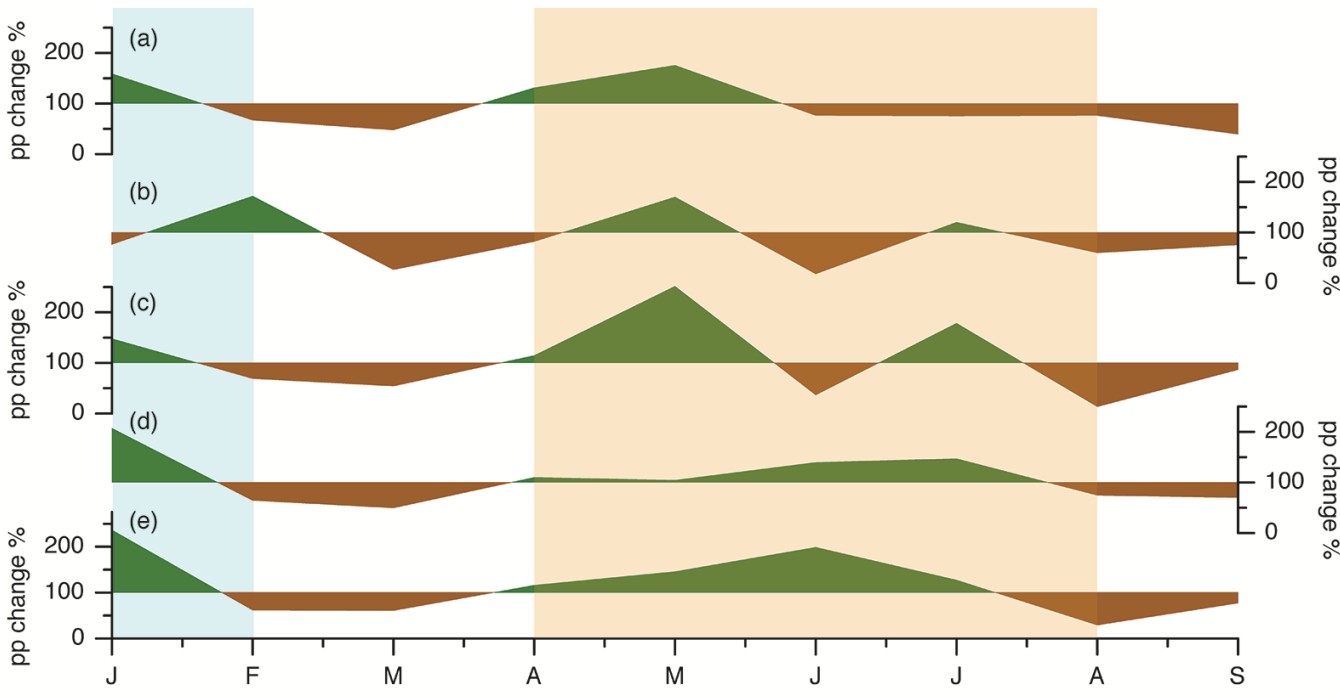

**Figure 8. Monthly precipitation from January to September 2019 at a) Scărişoara Ice Cave (ROU), b) Velika ledena jama v Paradani (SLO), c) Crna Ledenica (CRO), d) Vihren (BG) and e) Chionotrypa (Falakro, GRE). The data is shown in percentage deviation from the 1971-2000 average (represented by the 100 % mark). Green shading on the charts indicates precipitation above average and brown shading indicates precipitation below average. The blue rectangle shows the period of ice accumulation and the orange rectangle the ice ablation period of the investigated glaciers.**

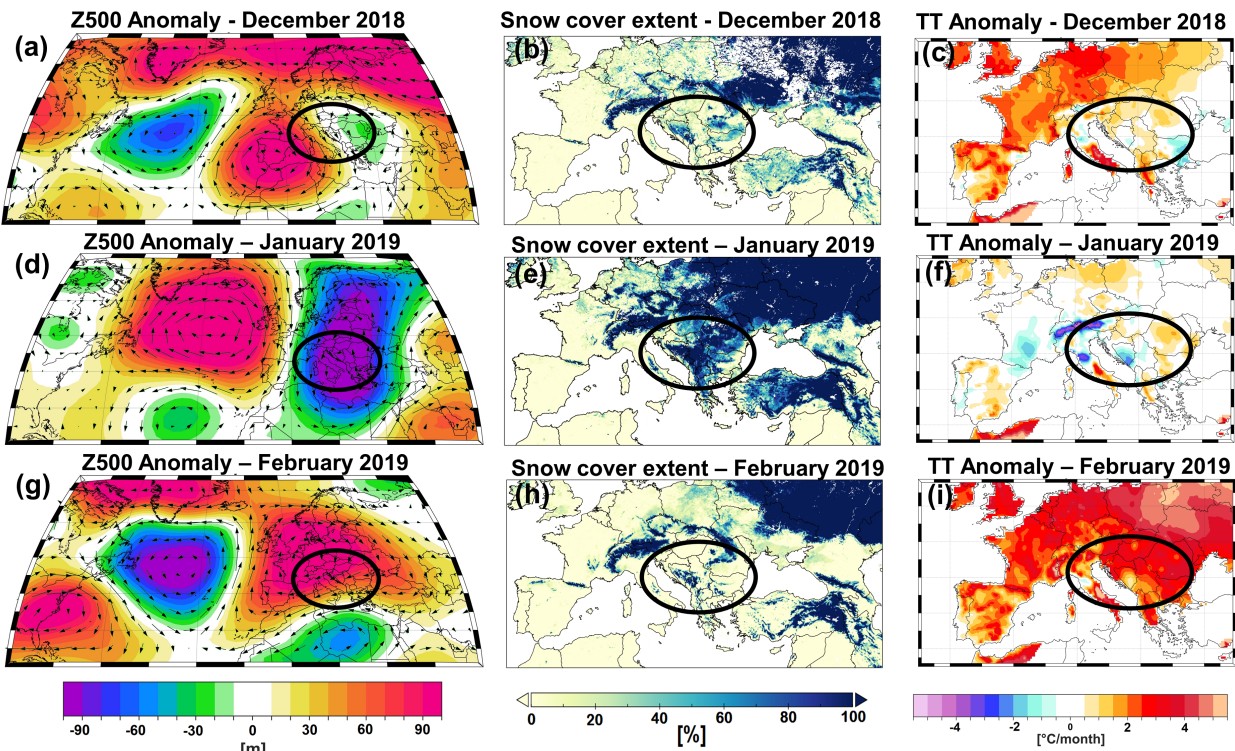

**Figure 9. Left column: geopotential height anomalies at 500mb level (Z500) [a) December 2018; d) January 2019 and g) February 2019]. Middle column: snow cover extent across Europe [b) December 2018; e) January 2019 and h) February 2019]. Right column: mean air temperature anomalies [c) December 2018; f) January 2019 and i) February 2019]. For all analyzed variables the anomalies are computed relative to the climatological period 1971 – 2000. Black ellipses indicate the study area.**

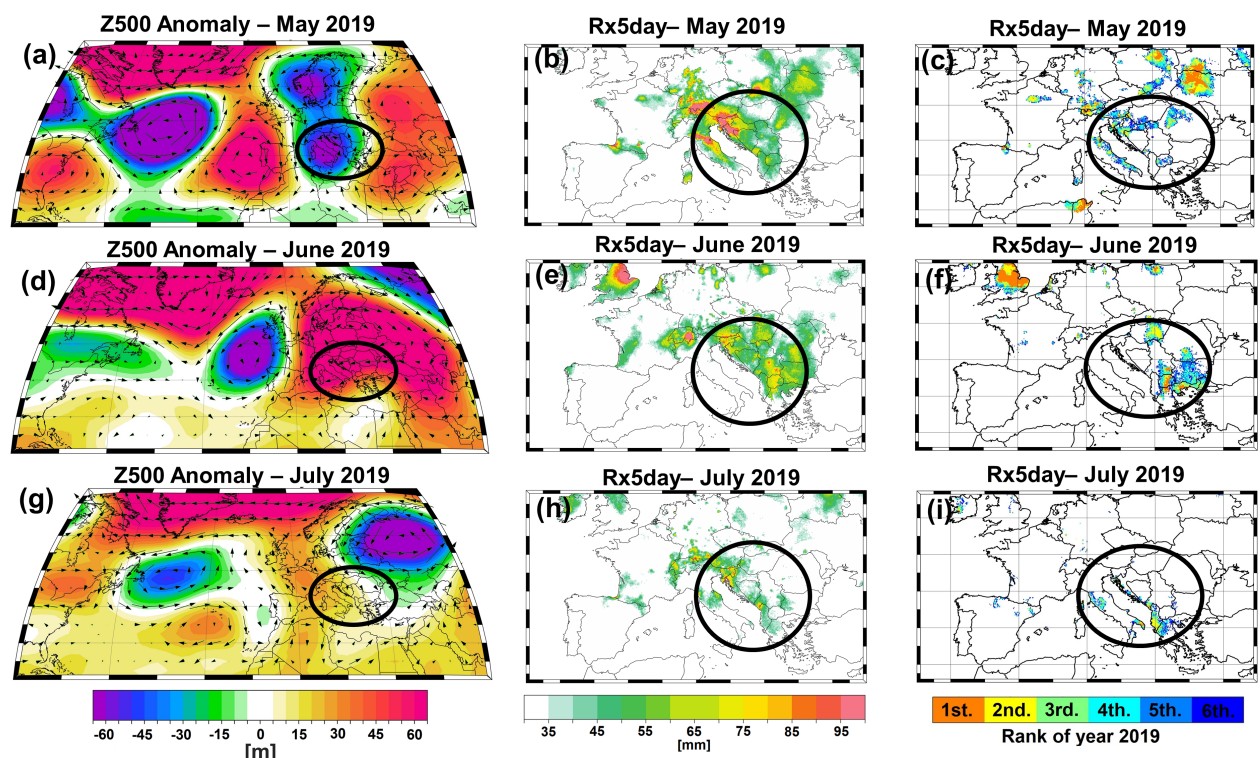

670 Figure 10. Left column: geopotential height anomalies at 500mb level (Z500) [a) May 2019; d) June 2019 and g) July 2019]. Middle column: highest 5-day precipitation amount (RX5day) [b) May 2019; e) June 2019 and h) July 2019]. Right column: top-six ranking of 2019 monthly RX5day [c) May 2019; f) June 2019 and i) July 2019]. A score of 1 represents the wettest month since 1950, a score of 2 represents the second wettest, and so on; all ranks greater than 6 are shown in white. Analyzed period: 1950–2015. For the Z500 the anomalies are computed relative to the period 1971 675 – 2000. Black ellipses indicate the study area.

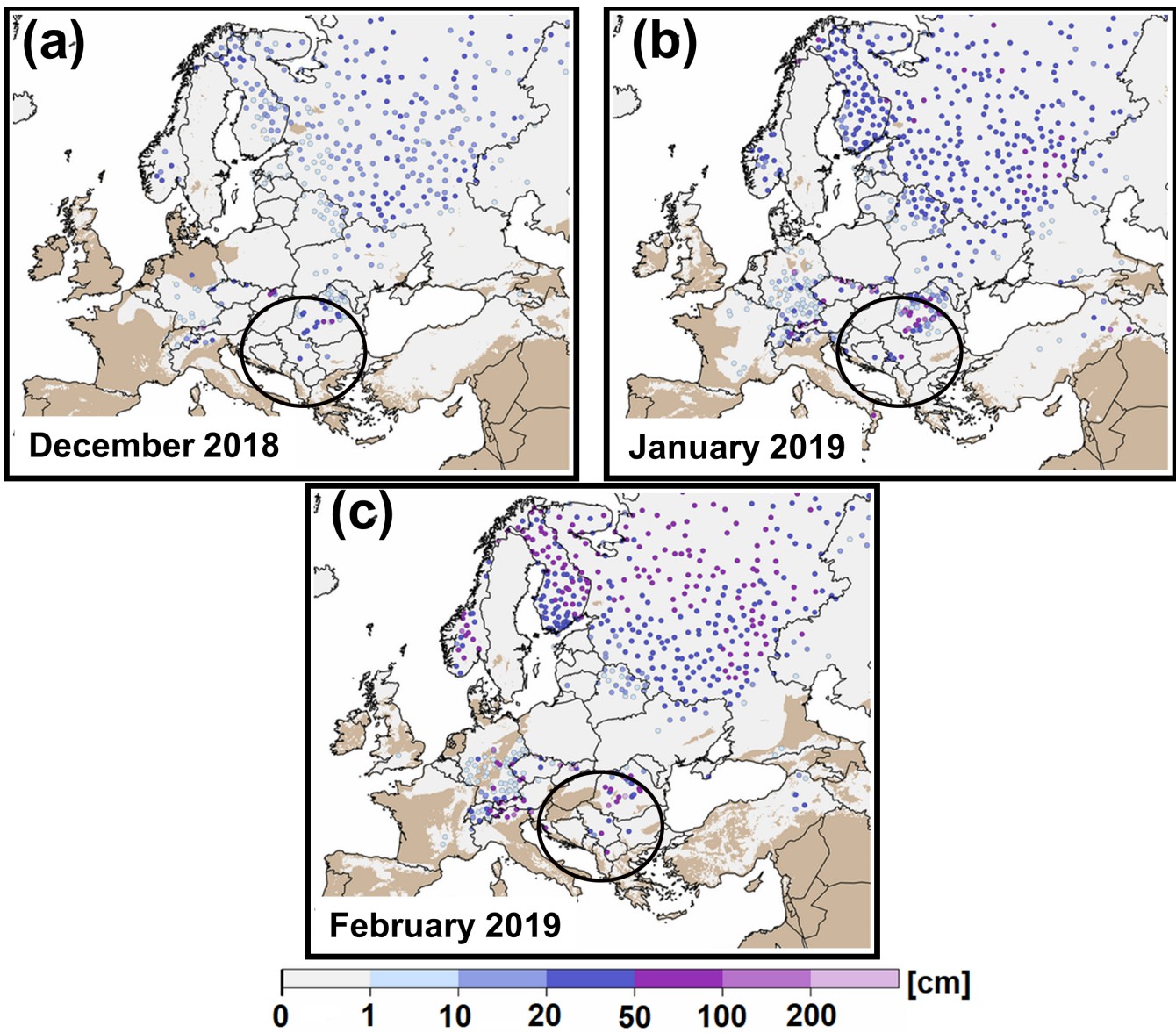

**Figure 11. Snow depth in Europe, winter 2018-2019. The black circle marks the study area. Snow depth data was provided by Deutscher Wetterdienst (www.dwd.de, last access on December 18, 2020).**

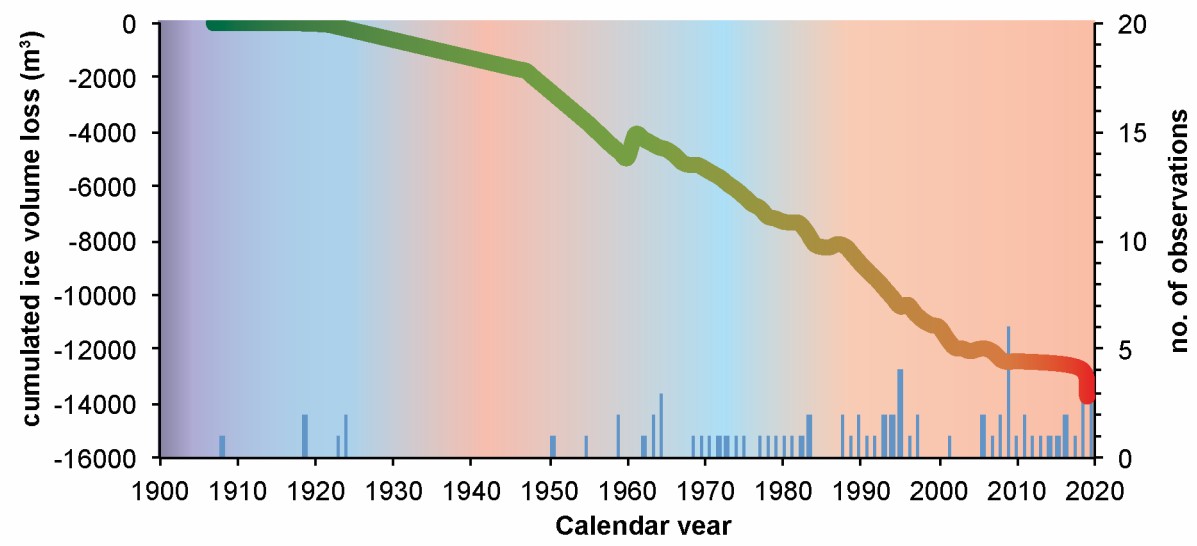

**Figure 12. Cumulative ice volume loss for European cave glaciers since 1900 (modified from Kern and Persoiu, 2013). Blue bars indicate the number of observations (caves) and the background colors show the average global temperature changes (red – warm, blue – cold).**

**Table 1. Uncertainties of ice level measurements and 2019 ice loss determination.**

| Ice Cave | Method | Measurement uncertainty in cm and % (percentage relative to 2019 ice loss) |
|---|---|---|
| Scărişoara | Ruler | 0.3 cm (<1 %) |
| Velika ledena jama v Paradani | Ruler | 0.3 cm (<1 %) |
| Chionotrypa Falakro | Photogrammetry | 30 cm (10 %) |
| Chionotrypa Olympos | Estimated | 50 cm (25 %) |
| Crna Ledenica | Photogrammetry | 20 cm (10 %) |

**Table 2. Characteristics of digital surface models and orthophotos obtained for Snezhnika and Banski Suhodol glacierets.**

| Glacieret | Date of UAV flight | DSM resolution (cm) | Ortophoto Resolution (cm) | Mean error DSM (cm) | Mean error Ortophoto (cm) |
|---|---|---|---|---|---|
| Snezhnika | 28.10.2018 | 5 | 2.1 | 1.5 | 0.2 |
| | 6.09.2019 | 22.5 | 2.8 | 2.1 | 0.3 |
| Banski Suhodol | 29.10.2018 | 10.6 | 4.9 | 1.9 | 0.3 |
| | 5.09.2019 | 18.7 | 4.6 | 1.6 | 0.2 |