# Peer review of "Record summer rains in 2019 led to massive loss of surface and cave ice in SE Europe"

_The Cryosphere, 2020_

## Referee Comment (RC1) · Anonymous Referee #1 · 24 Nov 2020

Overview of the manuscript

This manuscript investigates the changes of ice thickness of five ice-filled caves (Scarisoara, Chionotrypa-Falakro, Chionotrypa-Olympus, Crna Ledenica and Velika ledena jama v Paradani), as well as the area changes of two mountain glaciers (Snezhnika and Basnki Suhodol), all of them in Eastern Europe, during the hydrological year 2018-2019. The relatively large changes observed are associated to an anomaly in the weather (both summer and winter weather). The observations of ice changes are carried out based on in situ length measurements for the ice within caves, and with a drone for the two mountain glaciers. Weather parameters are obtained from the fol-

lowing datasets: E-OBS (Cornes et al., 2018), NCEP/NCAR (Kalnay et al., 1996) and MODIS/Terra Snow Cover Monthly L3 Global (Hall et al., 2006).

The manuscript is well-written and illustrates that changes in weather have also effects on ice within caves. However, in my opinion, there are some major flaws that need to be improved before this manuscript is suited for publication. Below a list of general comments and specific comments for the authors in order to improve the manuscript:

General comments:

1 – Weather vs climate:

The abstract, introduction, discussion and conclusions manuscript refer in numerous occasions to "climate". This, in my opinion, is not correct, since a study of one-year length does not reflect climate variability, and contains the high-frequency effects of the weather. Similarly, it is also erroneous to associate the observed changes of one year to climate. To "filter" the weather from the climate, the study should span minimum 10 years (e.g. Marzeion et al., 2014). Through the manuscript this should be clarified, and the word "climate" should be used minimally. Also, the areas of interest are located at really large distances within each other (up to 1000 km apart), and it is almost certain that each area of interest will have a different response to weather and climate. It can be expected that the glacier changes are not simply explained from pressure and temperature anomalies. For example, the two studied glaciers are still existing likely due to their elevation range, slope and aspect.

2 – Surface vs cave glacier:

The manuscript shows observations on two glaciarized systems: mountain glaciers and ice within caves. These two categories are rather blurry throughout the manuscript. Each type of ice is measured with different methods, but throughout the manuscript there is no distinction of them, and all the results and discussion are presented regardless of this difference. A distinction between both types of ice would make, in my

opinion, a cleaner section of methods and results, showing (1) changes in ice within caves, with its associated method and (2) changes in mountain glaciers from UAV. Finally, in connection to the general comment nr1, the two glaciarized systems will likely have different responses to weather and should not be discussed without taking this into account.

3 – Data collected:

The manuscript uses two main kinds of observations collected in situ: (1) distance measurement between benchmarks and the ice, to measure relative changes in the level of ice in caves, and (2) comparison of UAV-based surveys. These observations lack on specific description of how they are carried out. See specific comments on questions that arise when reading the manuscript. In general, since there are rather limited studies of ice within caves, it would be worth adding some paragraphs, even some photographic material, on how these measurements are conducted.

The temporal resolution of the data collected in caves is also not clear. Are the measurements done only once a year? In this case, the temporal resolution of the data collected does not allow making statements regarding the seasonal variability of the ice, as is stated throughout the discussion.

A table showing an overview of the data collected would be really beneficial for the reader to understand the amount and main characteristics of observations done in this study.

4 – Results:

The manuscript is also lacking methodological information on how some results are calculated, which limits the study's repeatability. For example, how is the change in the level of the ice converted to a volume change?

None of the presented results contains error bars. The general reader has no sense of the robustness of the measurements (also due to the lack of description in the data

collected), and this, in my opinion, causes a lack in scientific rigor of the results.

The section 4.1 (Ice mass balance changes) does not show any mass balance number. Neither volume change (presented for ice within caves) nor area changes (presented for mountain glaciers) is "mass balance" (Cogley et al., 2011).

The methods show that the UAV-based surveys produced Digital Elevation Models (DEMs), but these were not included in the study. Differencing of UAV-based DEMs is a robust method to infer volume changes and mass balance (e.g. Whitehead et al., 2013, Groos et al., 2019). Analyzing area changes in mountain glaciers, as opposed to volume changes, is not optimal, since the area changes are not as closely connected to climate/weather than the volume changes (e.g. Jóhannesson et al., 1989).

Specific comments

Title: "Unprecedented" is a very strong statement and the limited observations do not prove whether or not there has been any similar event in the past.

Title: I found confusing the term "loss of surface and cave ice". Something like "loss of ice in mountain glaciers and within caves" might be clearer.

L19-24: Half of the abstract is focused on "climate", but this manuscript does not show "climate" but "weather" (general comment nr 1).

L26: "catastrophic and unprecedented" ... again this is a really strong statement without clear evidences of it.

L26-30: The second half of the abstract is focused on model predictions and the fate of the ice within caves, but this manuscript does not show any climate model prediction at any point. Similarly, the paleoclimatic information is only mentioned once in the introduction (L50). Since the focus of this manuscript consists of bringing observations and exploiting the weather datasets, some general results should be mentioned in the abstract.

L32-40: See general comment nr 1.

L87-88: "Ice dynamics" typically refers to ice motion and ice deformation. As I understand this is not measured in this case.

L125: This study does not show "mass balance changes" (general comment nr. 4)

L127-130: How are the distance measurement carried out? With tape, total station? What's the estimated uncertainty of the measurements? (general comments nr. 3 & 4.)

L136-142: More details are needed for the photogrammetric set up. Did you use GCPs and/or GPS? How are the results of the photogrammetric processing, for example from bundle block adjustment? What is the expected uncertainty of the orthomosaic and the DEM? What's the exact date of survey? (see suggestion of adding a table with observations). Why are the DEMs not used in the study? Measuring elevation difference and volume changes is much more representative to study glacier changes than measuring area changes, since the area changes are influenced by the response time of the glacier (general comment nr. 4).

L144-145: "In order to link the (…) parameter (…)" The study does not perform any robust link or correlation between parameters. Please rephrase or clarify.

L143-154: Some information about the uncertainties of the weather parameters would also be highly valuable.

L156: See general comment nr. 4. No mass balance changes are provided in the results. Please use more accurate terms, such as "volume changes", "ice-level changes" or "area changes".

L163: How is this volume calculated from the ice-level changes? What are the uncertainties? This also applies for the other caves (general comment nr. 4)

L164-165: "a gradual decrease of the ice volume was evident since 2014, reaching a

minimum in September 2019 (Fig. 3)": This statement needs a stronger support than Fig. 3.

L242-244: "(...) resulted in the large accumulation of snow (...)" ... but this is only observed by the weather datasets, right? And these datasets do not show accumulation (snow thickness), only snow distribution. Therefore, this sentence might not be correct.

L245: "led to the rapid melt of the surface snowpack" ... again this is not observed, only suggested. Please rephrase acknowledging the lack of such specific observations, for example "high temperatures suggest rapid melting of the surface snowpack".

L248: "resulted in rapid ice accretion in caves" ... Please rephrase acknowledging the lack of such specific observations

L250: "wet late spring and summer led to rapid cave ice ablation" ... Please rephrase acknowledging the lack of such specific observations. Check for any other occurrences throughout the discussion.

L264: Break into a new paragraph, since now you start talking about mountain glaciers as opposed to ice within caves.

L271: Please follow a logical structure of the discussion, I suggest first a discussion about ice within caves and then a discussion about mountain glaciers, but not an alternation between the two.

L298: How is this prediction done? This shouldn't be stated in the conclusions without specifying any prediction of disappearance throughout the manuscript.

L302: "our observations show..." These are not really observations done in the study, these are results from weather datasets. To me, the observations done in this study are the ice level changes and area changes.

L303-310: Again, this is the first time when the prediction of extreme weather is mentioned in the manuscript. This should not be presented in the conclusions.

Figures

Fig. 2: Please indicate source of data in the caption.

Fig. 3: This figure showing changes in snow is heavily influenced by the seasonal differences of each picture. This needs to be properly addressed or otherwise this figure should not be presented. Comments on the caption: the figure does not show any ice. Also, the first year is 2014 and not 2016.

Fig. 4: See general comment nr. 4 about the limited use of the UAV data. Comments in the caption: "Orthomosaics showing ice surface changes (. . .)"

Fig. 5: See general comment nr. 1 and 2. The interpretation of pp change at several locations at relatively large distances and with different systems (mountain glacier vs caves with ice) is not straight-forward and it can be misleading to compare them as such.

Fig. 6: Letters (a,b...i) are missing.

Fig. 8: See general comment nr. 4. Volume changes is not the same as mass balance.

References

Cogley, J.G., R. Hock, L.A. Rasmussen, A.A. Arendt, A. Bauder, R.J. Braithwaite, P. Jansson, G. Kaser, M. Möller, L. Nicholson and M. Zemp: Glossary of Glacier Mass Balance and Related Terms, IHP-VII Technical Documents in Hydrology No. 86, IACS Contribution No. 2, UNESCO-IHP, Paris, 2011.

Cornes, R., Van der Schrier, G., Van den Besselaar, E. J. M., and Jones, P. D.: An Ensemble Version of the E-OBS Temperature and Precipitation Datasets, J. Geophys. Res., 123, 9391–9409, https://doi.org/10.1029/2017JD028200, 2018.

Groos, A.R.; Bertschinger, T.J.; Kummer, C.M.; Erlwein, S.; Munz, L.; Philipp, A. The Potential of Low-Cost UAVs and Open-Source Photogrammetry Software for High-Resolution Monitoring of Alpine Glaciers: A Case Study from the Kanderfirn (Swiss

Alps). Geosciences, 9, 356, 2019.

Hall, D. K. and Riggs, G. A.: MODIS/Terra Snow Cover Monthly L3 Global 0.05Deg CMG; Version 6; NASA National Snow and Ice Data Center Distributed Active Archive Center: Boulder, Co, USA, 2006.

Jóhannesson, T., Raymond, C. & Waddington, E. Time-scale for adjustment of glaciers to changes in mass balance. Journal of Glaciology 35, 355-369, 1989.

Kalnay, E., Kanamitsu, M., Kistler, R., Collins, W., Deaven, D., Gandin, L., Iredell, M., Saha, S., White, G.,Woollen, J., Zhu, Y., Leetmaa, A., Reynolds, R., Chelliah, M., Ebisuzaki,W., Higgins, W., Janowiak, J., Mo, K. C., Ropelewski, C.,Wang, J., Jenne, R., and Joseph, D.: The NCEP/NCAR 40-year reanalysis project, B. Am. Meteorol. Soc., 77, 437–470, https://doi.org/10.1175/1520-0477(1996)077<0437:TNYRP>2.0.CO;2, 1996.

Marzeion B, Cogley JG, Richter K, Parkes D. Glaciers. Attribution of global glacier mass loss to anthropogenic and natural causes. Science. 345(6199):919-21. doi: 10.1126/science.1254702, 2014

Whitehead, K., Moorman, B. J., and Hugenholtz, C. H.: Brief Communication: Low-cost, on-demand aerial photogrammetry for glaciological measurement, The Cryosphere, 7, 1879–1884, https://doi.org/10.5194/tc-7-1879-2013, 2013.

---

## Referee Comment (RC2) · Anonymous Referee #2 · 6 Dec 2020

Overview and general comments: I have read the manuscript entitled "Unprecedented loss of surface and cave ice in SE Europe related to record summer rains in 2019". In the manuscript, the authors investigate the ice changes in 5 subsurface glaciers (ice-caves) and 2 of the southernmost European glaciers during 2018-2019 year. Ice measurements in caves were performed using usual methods for ice body changes in ice caves by fixed marks/points in walls and photogrammetry, while glacier changes were evaluated using a drone. The weather during 2018-2019 period was evaluated using E-OBS dataset. The authors conclude that the observed ice loss during 2018-2019 was caused by extreme rainfalls during spring-summer. This ice melt event was an unprecedented event, not recorded during the last century. The manuscript is inter-

esting to understand the response of ice caves and glaciers to extreme rainfall events. Some minor observations and suggestions must be solved before publication.

Line 1: Title. The term "unprecedented" should be accompanied by a temporal term e.g ("Unprecedented loss of surface and cave ice in SE Europe since the last century. . ..or 100-years")

Lines 87 and 94: appear two times "Chionotrypa cave", I guess the caves have the same name, but maybe the authors could write directly the second name attributed "Chionotrypa (Falakro)" and "Chionotrypa (Olympus)" or, to shorten it, "Chionortypa-F" "Chionotrypa-O" to avoid repeating the name, as it appears along the text. An acronym for "Velika Iedena jama v Paradani" is also recommended or for short like "Velika" or "Paradini"

Line 104: change "metamorphosed" by "transformation"

Line 126: The authors indicate that photogrammetry was carried out. In which caves was performed? Maybe the authors should add some additional information about the methods of photogrammetry, given that at the moment is not a usual method applied on ice caves, however, it is a powerful technique to know ice volume changes in caves.

Line 136-142: A table could be helpful showing the resolution, error of MDT generation, and the accumulated error when comparing both models.

Line 144: I have asked an expert colleague in meteorology and he suggests some changes and modifications in relation to the use of the E-OBS dataset.

- One of the main problems of this dataset is related to the extreme events. The resolution of E-OBS is 0.25x0.25, which for mountain areas and rainfall events is too low. The following database has been intensively tested:

https://www.ecmwf.int/en/forecasts/datasets/reanalysis-datasets/era5

Line 167: Is it possible to indicate the drop or change in cm or m? Following a similar

descriptions of the rest of ice caves where authors describe the drop of ice in cm or m, and then the corresponding volume.

Line 198: Why did the authors use the 1971-2000 period instead of the last one (1980-2010)?

Line 307 "generate (semi)quantitative" add space.

Figures

Figure 1: Add legend, Red stars: ice caves. Blue stars: glaciers

Figure 2: It would be nice to show all ice changes from the caves of the study in the graph maybe in a zoomed area.

Figure 4: The photos should be labelled as a). b) etc

Figure 5: The photos should be labelled with a code for a quick identification. Maybe left= 1 and corresponding photos 1a, 1b, 1c, Middle= 2a, 2b, 2c. . .etc. and similar for Figure 7.

Figure 8: indicate the meaning of background colors (warm, cold...).

---

## Referee Comment (RC3) · Zoltán Kern (Referee) · 7 Dec 2020

Dear Authors, Dear Editor,

The manuscript reports enormous mass loss in five cave ice deposits and two small glaciers from SE Europe. Following the detailed evaluation of the meteorological conditions the Authors conclude that extreme precipitation events occurring in summer of 2019 led to catastrophic loss of ice. The study is well-structured and easy to follow; however, some revision can be recommended. I encourage the authors for the revision because the paper has the potential to became a key reference in the field of ice cave science.

[Figure]

best wishes, Zoltan Kern

General comments:

-I've found a bit confusing the usage of the term "glacier" in the paper. sometimes it seems that the Authors include both surface and cave glaciers in this term sometimes only surface glaciers. I suggest using the term "glacieret" when speaking about Snezhnika and Sudohol. It might help to avoid confusion.

-I missed two highly relevant references from the discussion (Colucci et al., 2016, Colucci and Gugliemlin 2019), and suggested additional references or replacing the currently cited reference with a more pertinent one at a couple of places. Colucci RR, Fontana D, Forte E, et al. (2016) Response of ice caves to weather extremes in the Southeastern Alps, Europe. Geomorphology 261: 1–11. Colucci, R.R., Guglielmin, M. 2019 Climate change and rapid ice melt: Suggestions from abrupt permafrost degradation and ice melting in an alpine ice cave. Progress in Physical Geography 43: 561-573 https://doi.org/10.1177/0309133319846056

-The name of one of the studied ice bodies is frequently written with a spelling mistake. "Basnki" should be corrected to "Banski" e.g. in line 114, 262, 266 or Fig4. I've marked this spelling mistake where I realized in the annotated PDF. I will not list them among the specific comments.

Although I marked few potential typos in the PDF, I note that I cannot provide a detailed linguistic review since I'm not a native English speaker.

specific comments:

line52: Citing this book chapter is not really relevant here, maybe the Authors wish to cite from the same year from the same author this paper: DOI:10.1017/RDC.2018.96

line62: A recent evaluation of CMIP5 simulations for this region (https://doi.org/10.1007/s00704-020-03361-7) confirmed the increase in winter precipitation, however showed that the models which reproduced better the decadal

hydroclimate variability of the 1850-2005 period show less reduction in summer precipitation (instead increase). This suggests that it is more likely that summer precipitation will not decrease so highlight the importance of the topic of the study. It might be a relevant info here.

line68: I think Kern and Thomas 2014 is a more pertinent citation for this statement than Kern et al., 2013

line76: Eisreisenwelt is not counted among the top5? If so then please, revise this statement.

line85: Is it really 'equilibrium' or just a moderately negative mass balance?

line98: Geographical coordinates are lacking for Crna Ledenica.

lines187-188: Are the described changes (retreat at the lower end of the two glacierets, increase in the width of the rimaye separating them from the cirque headwall) visible in Fig4? If yes, please refer to the figure.

line238: Which zero isotherm do you mean? MAAT, Summer meant, July mean T? Please, be more specific. line238-241: I think a supporting reference for this statement is needed.

lines276-278: for instance, this is a place where Colucci et al., 2016, Colucci and Gugliemlin 2019 could be incorporated to the discussion.

Some additional minor comments are marked in the annotated PDF.

Figures

Fig2: decimal places in the y-axis values can be omitted.

Fig3: The caption says that the changes are illustrated "since 2016", however the date of the first photo is 10.08.2014. Please, clarify this.

Fig6: I cannot see the panel codes in the figure. In addition, please, increase the

[Figure]

characters in the title of the maps and use uniform character size for the titles.

Fig7: I cannot see the panel codes in the figure. In addition, please, increase the characters in the title of the maps and use uniform character size for the titles.

Fig8: Comment1: Could you add a color scale to help a better interpretation of the illustration. For instance, I guess reddish colors show warmer temp and bluish colors show colder temp, however it should be indicated in the caption. Comment2: Why the global temperature? European or SE European temperature changes could be much more relevant.

Please also note the supplement to this comment:
https://tc.copernicus.org/preprints/tc-2020-287/tc-2020-287-RC3-supplement.pdf

[Figure]

**Supplement:**

[revised manuscript text omitted]

---

## Author Comment (AC1) · 3 Jan 2021

We thank the reviewer for the time dedicated to our manuscript and the detailed comments. We addressed these in the attached file (RC comments in red, AC in black).

Please also note the supplement to this comment:
https://tc.copernicus.org/preprints/tc-2020-287/tc-2020-287-AC1-supplement.pdf
* * *

---

## Author Response (AR1)

We thank the reviewer for the detailed consideration of our manuscript. We believe that the glaciological perspective emphasized in the review complements our (mainly) speleological one and the revised manuscript (in accordance with the comments) clarifies the contended issues.

We have entirely rewritten the "methods" section, added new information under "results" (including several new figures) and expanded the "discussions" section. Detailed responses, including the modifications of the manuscript can be found in the attached file (RC comments in red, AC in black).

Overview of the manuscript

This manuscript investigates the changes of ice thickness of five ice-filled caves (Scarisoara, Chionotrypa-Falakro, Chionotrypa-Olympus, Crna Ledenica and Velika ledena jama v Paradani), as well as the area changes of two mountain glaciers (Snezhnika and Basnki Suhodol), all of them in Eastern Europe, during the hydrological year 2018-2019. The relatively large changes observed are associated to an anomaly in the weather (both summer and winter weather). The observations of ice changes are carried out based on in situ length measurements for the ice within caves, and with a drone for the two mountain glaciers. Weather parameters are obtained from the following datasets: E-OBS (Cornes et al., 2018), NCEP/NCAR (Kalnay et al., 1996) and MODIS/Terra Snow Cover Monthly L3 Global (Hall et al., 2006).

The manuscript is well-written and illustrates that changes in weather have also effects on ice within caves. However, in my opinion, there are some major flaws that need to be improved before this manuscript is suited for publication. Below a list of general comments and specific comments for the authors in order to improve the manuscript:

General comments:

1 – Weather vs climate:

The abstract, introduction, discussion and conclusions manuscript refer in numerous occasions to "climate". This, in my opinion, is not correct, since a study of one-year length does not reflect climate variability, and contains the high-frequency effects of the weather. Similarly, it is also erroneous to associate the observed changes of one year to climate. To "filter" the weather from the climate, the study should span minimum 10 years (e.g. Marzeion et al., 2014). Through the manuscript this should be clarified, and the word "climate" should be used minimally.

In most of the cases, the usage of the word "climate" instead of "weather" is a carryover from loose oral communication. Where this was the case, we made the necessary amendments. In some cases, though, we kept the word climate" as we referred to "climate". We did so in parts of the introduction and discussions/conclusions, where we placed the observations of interactions between weather and glacial processes (cave and surface) during 2019 into a long-term, context ("climate"). We also make a point here on surface *vs.* cave weather/climate: given the particular conditions in caves – long-term stability of air temperature, reduced exchange with the external atmosphere (mainly through conduction and with minimal convection/advection etc), caves do not exhibit variability of meteorological elements that could be defined as „weather", they only have climate. This has important implications for the dynamics of glacial processes in caves, with external outside extreme events (like the ones investigated in this paper) playing an outsized role in both the mass and volume changes of perennial ice accumulations. We have explained this better in our revised manuscript, both in the introduction and in the discussions.

To "filter" the weather from the climate, the study should span minimum 10 years (e.g. Marzeion et al., 2014).

This is something we were doing for a while now for selected ice caves and started for the surface glaciers. However, the focus of this paper is not to discuss the long-term changes of cave ice mass and volume, but to present and discuss the role of extreme precipitation events on ice mass/volume changes and how short-term weather variability can impact small glaciers (as opposed to long-term climate variability). We introduced a new paragraph to better convey this and also expanded the discussion accordingly:

Whereas ice loss in surface glaciers is mostly due to melting related to rising temperatures (e.g., Marzeion et al., 2014), cave ice ablation is primarily due to drip water delivering heat to the ice (Luetscher et al., 2005; Persoiu et al., 2011a; Colucci et al., 2016). Therefore, whereas the projected increase in air temperature in mountain areas would result in enhanced mass loss for surface glaciers, the same rising temperatures might only marginally affect ice mass balance in caves. Monitoring studies in ice caves has been done sporadically since the mid 20[th] century (Racovita, 1994; Luetscher et al., 2005; Persoiu and Pazdur, 2011; Kern and Persoiu, 2013), the results showing that reduction of winter precipitation and increase of winter temperatures are the main factors behind loss of ice, with summer temperatures having a negligible role.

Also, the areas of interest are located at really large distances within each other (up to 1000 km apart), and it is almost certain that each area of interest will have a different response to weather and climate. It can be expected that the glacier changes are not simply explained from pressure and temperature anomalies. For example, the two studied glaciers are still existing likely due to their elevation range, slope and aspect.

Our analyses of weather conditions have shown that the extreme events of 2019 had a consistent behavior over the entire investigated area (encompassed by a circle with a radius of about 400 km). While we do not expect that the investigated phenomena occurred simultaneously, they did occur *in all* studied places, with *similar characteristics* (intensity, duration) and affected *all* investigated caves. Abrupt reduction of ice levels (likely indicating negative mass balance) has also been

documented by us in several other caves in the region with anomalous weather conditions. Contrary, ice caves outside this area did not exhibit similar reductions in ice level/mass. However, in the absence of precise and accurate measurements, we did not include these observations in our analyses. Next, we did not explain the glacier changes as resulting from pressure and temperature anomalies, but as a result of massive addition of warm waters directly to the ice bodies. These high amounts of water delivered in a short time (coupled with the high specific heat of water) were responsible for the massive loss of ice in 2019 and one of our main messages is that extreme hydrological events have and will have an important role in the future evolution of (especially) small perennial surface and cave glaciers. This point has been strengthened in our text by referencing similar observations from both cave and surface glaciers throughout Europe. We also note that our work concentrates on the loss of ice during 2019 melt season, and we present weather data from the previous winter only to introduce a wider context for the ablation processes on which our manuscript focuses.

2 – Surface vs cave glacier:
The manuscript shows observations on two glaciarized systems: mountain glaciers and ice within caves. These two categories are rather blurry throughout the manuscript. Each type of ice is measured with different methods, but throughout the manuscript there is no distinction of them, and all the results and discussion are presented regardless of this difference. A distinction between both types of ice would make, in my opinion, a cleaner section of methods and results, showing (1) changes in ice within caves, with its associated method and (2) changes in mountain glaciers from UAV. Finally, in connection to the general comment nr1, the two glaciarized systems will likely have different responses to weather and should not be discussed without taking this into account.
Well, the surface and cave glaciers analyzed here are more similar to each other, than dissimilar. First, all cave glaciers are located below the cave entrances where external weather has a strong impact on ice behavior. This is valid mostly for precipitation and not so for air temperature (see also our comment on "general comment 1" above): snowfall and liquid precipitation directly reaches the ice and snow masses in caves. To illustrate this point, we have added a new figure with cross sections of the studied caves showing the relationship between underground ice/snow and outside environment. It is evident that the surface and cave ice accumulations have similar positions with respect to water input and thus are expected to respond similarly. Nevertheless, in our presentation of the results, we have clearly separated the analyses and we only discussed surface and cave glaciers together when their common behavior made such an analyses meaningful. In the revised manuscript we have clarified this point, by identifying the similarities and differences in the response of surface glacierets and cave glaciers to weather and climate variability before analyzing these for the particular conditions in 2019.

3 – Data collected:
The manuscript uses two main kinds of observations collected in situ: (1) distance measurement between benchmarks and the ice, to measure relative changes in the level of ice in caves, and (2) comparison of UAV-based surveys. These observations lack on specific description of how they are carried out. See specific comments on questions that arise when reading the manuscript. In general, since there are rather limited studies of ice within caves, it would be worth adding some paragraphs, even some photographic material, on how these measurements are conducted.
Thank you for this point. We have developed some of the techniques to identify the different components of ice level changes in caves (surface vs. basal etc) and we sometimes overlook explaining these in detail, thus limiting the understanding of data. We have a new paragraph (complete with graphic illustration) detailing these methods.

The temporal resolution of the data collected in caves is also not clear. Are the measurements done only once a year? In this case, the temporal resolution of the data collected does not allow making statements regarding the seasonal variability of the ice, as is stated throughout the discussion.
A table showing an overview of the data collected would be really beneficial for the reader to understand the amount and main characteristics of observations done in this study.
We have clarified this in the revised version of the manuscript, by adding information on the temporal resolution of measurements for each caves (for some, this has already been mentioned in the text). Second, these caves have been monitored for quite a long time (e.g., 70+ yrs for Scărișoara) and we know (within weeks) the periods of ice level minima and maxima. Subsequently, we are visiting the caves several times/year near these thresholds moments and record the changes. We are clarifying this in the "methods" section. A table with the observations was also added to the article.

4 – Results:
The manuscript is also lacking methodological information on how some results are calculated, which limits the study's repeatability. For example, how is the change in the level of the ice converted to a volume change?
None of the presented results contains error bars. The general reader has no sense of the robustness of the measurements (also due to the lack of description in the data collected), and this, in my opinion, causes a lack in scientific rigor of the results.
We have augmented the description of the methods used in our investigation as detailed also in the comment above. For some of the methods, we considered that a description is not necessary (e.g., to convert ice level changes in volume change,

we multiplied the thickness of the melted ice by the surface of ice over which melting occurred), while for the others, detailed descriptions were inserted in the text, together with a new explanatory figure and a table with the measurement errors.

The section 4.1 (Ice mass balance changes) does not show any mass balance number. Neither volume change (presented for ice within caves) nor area changes (presented for mountain glaciers) is "mass balance" (Cogley et al., 2011).
Yes, this is a fundamental question – what is the dimension of "mass balance", mass or volume (Cogley et al., 2011)? We settled for the later, perhaps unconsciously, as in most cases ice in caves forms by the freezing of water, hence attaining maximum density with extremely limited future density changes (the average density. Thus, we have used "mass balance" to mean "volume change". We do realize that this is not always the case (for example in the upper parts of the cave ice deposits formed of compacted snow) so we have modified the text accordingly. We also emphasize that we considered "mass balance" (and now volume changes) as a sum, not a rate (i.e., change over time, as this was not the aim of our study). Further, due to the restricted space in which cave glaciers are located (rock walls), potential mass balance changes induced by ice flow (i.e, non-climate related mass balance changes) are virtually inexistent. Where ice flow and basal melting does occur though (Scărișoara ice Cave), we have explained in the "methods" section how we dealt with it and the data reported in the "results" section consider only changes at the surface (i.e, climate-related ones).

The methods show that the UAV-based surveys produced Digital Elevation Models (DEMs), but these were not included in the study. Differencing of UAV-based DEMs is a robust method to infer volume changes and mass balance (e.g. Whitehead et al., 2013, Groos et al., 2019). Analyzing area changes in mountain glaciers, as opposed to volume changes, is not optimal, since the area changes are not as closely connected to climate/weather than the volume changes (e.g. Jóhannesson et al., 1989).
We did indeed produce DEMs, but decided not to use them; instead, we presented ortophotomosaics. In the revised manuscript, we added the two DEMs, as well as two figures showing the lowering of the surface of the glacierets. The text below was also added to the manuscript.
We note thought that in the case of the two investigated glacierets, the significant loss of glacier-covered area is a clear indicator of the impact of extreme events in summer 2019. This is also in accordance with the main finding of Jóhannesson et al. (1989) cited above, showing that the response time of glaciers to changes in climate could be significantly less than the $10^2$-$10^3$ theoretically expected years (i.e., within the time span of our observed and inferred changes).

Between 2018 and 2019 the terminus of both glacierets show significant loss of ice. The mean retreat of the terminus was 3.4 m at Snezhnika and 5.8 m at Banski Suhodol (fig. 7). Fig. 7 c and d shows the difference in the elevation of ice surface between 2018 and 2019. In general, the difference in Snezhnika ice surface elevation between 2018 and 2019 range between 0.2 and 1 m, exceeding 1.2 m only in the south-western part. On the other hand, more than half of Banski Suhodol surface has lowered with more than 1 m due to surface melting.

Specific comments
Title: "Unprecedented" is a very strong statement and the limited observations do not prove whether or not there has been any similar event in the past.
We have modified the title to better reflect our data and message: "Record summer rains in 2019 led to massive loss of surface and cave ice in SE Europe"

Title: I found confusing the term "loss of surface and cave ice". Something like "loss of ice in mountain glaciers and within caves" might be clearer.
We have modified the title to better reflect our data and message

L19-24: Half of the abstract is focused on "climate", but this manuscript does not show "climate" but "weather" (general comment nr 1).
Please see the response to general comment 1.

L26: "catastrophic and unprecedented" ... again this is a really strong statement without clear evidences of it.
The melting we have observed in 2019 is unprecedented for varying time periods (between 99 and 20 years) for the different ice bodies we have investigated. Nevertheless, the loss of ice was higher than any such loss previously recorded for all caves (for separated periods, though). This differentiation was made clear in the text and the relevant line in the abstract now reads: "Our investigation shows that extreme precipitation events occurring between May and July 2019 led to loss of ice at levels higher than any recorded for all investigated sites."

L26-30: The second half of the abstract is focused on model predictions and the fate of the ice within caves, but this manuscript does not show any climate model prediction at any point. Similarly, the paleoclimatic information is only

mentioned once in the introduction (L50). Since the focus of this manuscript consists of bringing observations and exploiting the weather datasets, some general results should be mentioned in the abstract.
1. The entire final paragraph of the article is a discussion of model predictions of future extreme events. In summary we argue that 1) global warming and associated Arctic amplification will 2) lead to meridional amplification and slower propagation of the Rossby waves, thus 3) further leading to increased frequency of blocking conditions finally resulting in 4) in more frequent (and possibly stronger) extreme events. We support this with relevant literature citation and indicate that in the context of our findings showing high sensitivity of cave and surface glaciers in SE Europe to extreme precipitation, the model-predicted increase in the frequency and intensity of extreme events threatens the survival of these ice bodies. This is summarized in the abstract as "As climate models predict that such extreme precipitation events are set to increase in frequency and intensity, the presence of cave glaciers in SE Europe and the paleoclimatic information they host may be lost in the near future."

2. We have added a new paragraph summarizing our specific findings, as follows:
In this context, we present here the response of cave and surface glaciers in SE Europe to the extreme precipitation events occurring between May and July 2019 in SE Europe. Surface glaciers in the northern Balkan Peninsula lost between 17 and 19 % of their total area, while cave glaciers in Croatia, Greece, Romania and Slovenia lost ice at levels higher than any recorded by instrumental observations during the past decades. The melting was likely the result of large amount of warm water delivered directly on the surface of the glaciers leading to rapid reduction of ice covered area of surface glaciers and thickness of cave ones.

L32-40: See general comment nr 1.
See our response for general comment 1.

L87-88: "Ice dynamics" typically refers to ice motion and ice deformation. As I understand this is not measured in this case.
It is ice level changes.

L125: This study does not show "mass balance changes" (general comment nr. 4)
It is ice level/volume changes (see the detailed response to general comment 4)

L127-130: How are the distance measurement carried out? With tape, total station? What's the estimated uncertainty of the measurements? (general comments nr. 3 & 4.)
We have expanded the methods section to include this information and also included a new figure to show how ice levels are measured in caves. The relevant response from the main text is below:
The first set measurements were made with a measuring tape, and the second along a metal line embedded in ice. The precision was better than 0.3 mm in both cases.

L136-142: More details are needed for the photogrammetric set up. Did you use GCPs and/or GPS? How are the results of the photogrammetric processing, for example from bundle block adjustment? What is the expected uncertainty of the orthomosaic and the DEM? What's the exact date of survey? (see suggestion of adding a table with observations). Why are the DEMs not used in the study? Measuring elevation difference and volume changes is much more representative to study glacier changes than measuring area changes, since the area changes are influenced by the response time of the glacier (general comment nr. 4).
Details of the photogrammetric studies have been added (see the text below). Also, we have added a table with the following characteristics of the digital surface models and ortophotos: Date of UAV flight, DSM resolution (cm), Ortophoto Resolution (cm), Mean error DSM (cm), Mean error Ortophoto (cm)

The drone survey was designed to cover the glaciers and their surroundings. For flight planning and mission control we used DJI Ground Station Pro software. The flight height was set at 200 m and the images were collected every three seconds along parallel lines with an overlap of 80%. Among camera locations, ground control points (GCPs) were used for both glaciers (5 for Snezhnika glacier and 8 for Banski Suhodol glacier) to georeference the digital surface models and the orthophotos. The GCPs were measured with a high accuracy Hiper V Topcon real-time kinematic (RTK) positioning system. The images were processed in Agisoft Photoscan Professional by using the following workflow: 1. Alignment and match of photos at the highest accuracy → 2. Analysing the sparse cloud → 3. Importing and setting the GCPs on each camera and the free-network bundle adjustment → 4. Generating the referenced dense cloud with medium accuracy settings and moderate depth filter → 5. Generating a mesh and textures → 6. Create the Digital Surface Model → 7. Create the Orthophoto-mosaic. Using the above-mentioned workflow, high-resolution digital surface models were created for both glacierets

L144-145: "In order to link the (. . .) parameter (. . .)" The study does not perform any robust link or correlation between parameters. Please rephrase or clarify.

We have rephrased this to read: "In order to understand the role of large scale circulation patterns in determining specific weather types, we have computed […]"

L143-154: Some information about the uncertainties of the weather parameters would also be highly valuable.

More details about the uncertainties and the parameters employed in our study have been added in the methods section.

L156: See general comment nr. 4. No mass balance changes are provided in the results. Please use more accurate terms, such as "volume changes", "ice-level changes" or "area changes".

We amended the text to be more specific.

L163: How is this volume calculated from the ice-level changes? What are the uncertainties? This also applies for the other caves (general comment nr. 4)

We multiplied the thickness of melted ice layer with the surface over which this occurred. The combination of cave morphology and ice accumulation and ablation processes resulted in a perfectly flat upper surface of the ice block (Perşoiu et al., 2011, Perşoiu ad Pazdur, 2011, Perşoiu, 2018 etc).

We have included a table under the "methods" section, in which all relevant information is given, including separate uncertainties for measurements and calculations.

L164-165: "a gradual decrease of the ice volume was evident since 2014, reaching a minimum in September 2019 (Fig. 3)": This statement needs a stronger support than Fig. 3.

Unfortunately, we do not have data to offer a stronger support, but we believe that the observations taken at the end of the melting season and reflected in figure 3 are relevant.

L242-244: "(. . .) resulted in the large accumulation of snow (. . .)" . . . but this is only observed by the weather datasets, right? And these datasets do not show accumulation (snow thickness), only snow distribution. Therefore, this sentence might not be correct.

Because we focused our article on the summer ice loss, we did not include all relevant information of winter accumulation. We have now added data on snow thickness outside the caves. Because the cave ice deposits are directly fed by snowfall, this data is also relevant for the caves. Where ice forms by the freezing of water (Scărișoara ice Cave and Velika ledena jama v Paradani) we have added relevant ice level measurements.

L245: "led to the rapid melt of the surface snowpack" . . . again this is not observed, only suggested. Please rephrase acknowledging the lack of such specific observations, for example "high temperatures suggest rapid melting of the surface snowpack".

We did observed it, actually, as we were in the field, monitoring ice level changes in the two caves referenced here, Further, figures 6f and 6i show the rapid disappearance of surface snow in February 2019 and a strong positive temperature anomaly in the same month, respectively. We believe that the combination of field observation and meteorological data supports our text. We added a line specifically mentioning field observations in Chionotrypa Olympos, Chionotrypa Falakro and Crna Ledenica and measurements data in Scărișoara Ice Cave ("Ice level measurements indicate that a 15 cm thick layer of ice was added to the upper surface of the ice block in Scărișoara Ice Cave, exceeding the mean annual growth for the 2000-2018 period (Perşoiu and Pazdur, 2011, Perşoiu, 2018)")

L248: "resulted in rapid ice accretion in caves" ... Please rephrase acknowledging the lack of such specific observations

See above.

L250: "wet late spring and summer led to rapid cave ice ablation" . . . Please rephrase acknowledging the lack of such specific observations. Check for any other occurrences throughout the discussion.

We believe that here is a misunderstanding. This paragraph is based on the observations and measurements we have made in summer 2019 and are reporting here (data in the "results" section and in figs. 2 and 3).

L264: Break into a new paragraph, since now you start talking about mountain glaciers as opposed to ice within caves.

Perhaps the reviewer refers to line 261, where the discussion of the surface glacierets starts. Done.

L271: Please follow a logical structure of the discussion, I suggest first a discussion about ice within caves and then a discussion about mountain glaciers, but not an alternation between the two.

This paragraph was moved up to follow the discussion of ice caves.

L298: How is this prediction done? This shouldn't be stated in the conclusions without specifying any prediction of disappearance throughout the manuscript.

We added a new paragraph in the "discussions" section, where we expand our previous interpretation, see below

The dramatic increase in area loss, coupled with overall shallow thickness of these glaciers makes them especially vulnerable to rapid disintegration. The equilibrium line altitude for surface glaciers and glacierets in southeast Europe is well above the highest peaks (Hughes, 2018) so they are in a continuous ice mass loss, with extreme events like the one we have described threatening their survival. Extrapolating our data, episodes of rapid summer melt induced by similar extreme events, either heat waves (Hughes, 2008) or precipitation (this study) could result in the loss of surface ice in the coming decade. Similarly to the glacierets we investigated, perennial snow patches on Mt. Olympus, remnants of glaciers from the last glacial cycle (Styllas et al., 2018), began to disintegrate in 2019 under the prolonged heat wave. These cases mirror recent findings from SW Europe, where Moreno et al. (2020) have shown that glaciers surviving warm periods of the past 2000 years are rapidly melting, being at the risk of disappearing within the coming decade(s).

L302: "our observations show..." These are not really observations done in the study, these are results from weather datasets. To me, the observations done in this study are the ice level changes and area changes.

These are the results of our observations, measurements and data interpretation. We summarized them as "our results", see below: "While cave and surface glaciers in mountains across Europe are sensitive to increasing temperature, we showed here that extreme summer rains led to rapid melting and disintegration of ice bodies, rendering them even more sensitive to temperature changes."

L303-310: Again, this is the first time when the prediction of extreme weather is mentioned in the manuscript. This should not be presented in the conclusions.

Although not labeled as such, the following paragraph in the "discussions" section presents future changes in extreme weather in SE Europe – causes, manifestation and likely impact on glaciers.

All surface glaciers in southern Europe are out of balance with present-day climatic conditions, but the slow melting occurring at their termini results in gradual re-equilibration with local climatic conditions (Zekollari et al., 2020). However, recent rapid warming leads to an increase in the altitude of the 0 °C isotherm (Rottler et al., 2019) thus further increasing the imbalance between glaciers and climate and enhanced melting. Our results suggest that, adding to the melting under increased temperatures, heavy summer precipitation events result in enhanced melting of both cave and surface glaciers. With increasing temperatures, the altitudinal rise of the 0 °C isotherm (Rubel et al., 2017) would bring more glaciated terrain under warming conditions, and thus yet more susceptible to heat transfer during heavy summer thunderstorms and extreme summer heat waves. Accelerated warming of the Arctic (Holland and Bitz, 2003) would result in meridional amplification and slower propagation of the Rossby waves, leading to an increase in the frequency of blocking conditions and associated extreme events (Francis and Vavrus, 2012; Liu et al., 2012; Screen and Simonds, 2014). The increased frequency, duration and intensity of both heat waves (e.g., Spinoni et al., 2015) and heavy rainfall events (Púčik et al., 2017; Rädler et al., 2019) in southeast Europe would thus lead to a higher ablation rate of surface and cave glaciers than that expected from increased temperatures alone. Especially vulnerable are cave glaciers, already located in areas subject to both warming and extreme summer thunderstorms, and surface glaciers close to the 0 ° isotherm, thus resulting in the loss of ice faster than predicted by the most recent estimates (IPCC, 2019; Paul et al., 2020).

Figures

Fig. 2: Please indicate source of data in the caption.

Info added to the caption: Updated from Perşoiu and Pazdur (2011), The Cryosphere

Fig. 3: This figure showing changes in snow is heavily influenced by the seasonal differences of each picture. This needs to be properly addressed or otherwise this figure should not be presented. Comments on the caption: the figure does not show any ice. Also, the first year is 2014 and not 2016.

The pictures were taken at the end of the melting season. Depending on whether conditions, this could occur earlier or later in the year. Perhaps the resolution does not allow for ice to be visible in the photograph, but ice is there, nevertheless (check the left side of the deposit, where ice is exposed to a lesser degree in 2018 and higher in 2019, although not visible in the pictures taken in 2014 and 2016).

Fig. 4: See general comment nr. 4 about the limited use of the UAV data. Comments in the caption: "Orthomosaics showing ice surface changes (. . .)"

Caption updated. For the use of UAV data, see our response in reply to general comment 4

Fig. 5: See general comment nr. 1 and 2. The interpretation of pp change at several locations at relatively large distances and with different systems (mountain glacier vs caves with ice) is not straightforward and it can be misleading to compare them as such.
Ware sorry, but we do not understand this comment. The figure shows precipitation data at the studied locations in percentage deviation from the 1971-2000 average. The message, as conveyed by the green and brown shading, is that during summer 2019, precipitation amount values were extremely high (generally by 200 %) than the long term average. Second, the high precipitation values occurred during the period over which generally ice melting occurs in all caves (orange rectangle). Summarizing, the figure shows that in 2019 high precipitation amount were registered during the ablation period of cave and surface ice.

Fig. 6: Letters (a,b...i) are missing.
We added letters to all panels.

Fig. 8: See general comment nr. 4. Volume changes is not the same as mass balance.
The figure is an updated version of a previously published one by Kern and Perșoiu (2013) and we kept the original axes captions. We changed the vertical scale to read "cumulated ice volume loss (m3)".

References
Cogley, J.G., R. Hock, L.A. Rasmussen, A.A. Arendt, A. Bauder, R.J. Braithwaite, P. Jansson, G. Kaser, M. Möller, L. Nicholson and M. Zemp: Glossary of Glacier Mass Balance and Related Terms, IHP-VII Technical Documents in Hydrology No. 86, IACS Contribution No. 2, UNESCO-IHP, Paris, 2011.
Cornes, R., Van der Schrier, G., Van den Besselaar, E. J. M., and Jones, P. D.: An Ensemble Version of the E-OBS Temperature and Precipitation Datasets, J. Geophys. Res., 123, 9391–9409, https://doi.org/10.1029/2017JD028200, 2018.
Groos, A.R.; Bertschinger, T.J.; Kummer, C.M.; Erlwein, S.; Munz, L.; Philipp, A. The Potential of Low-Cost UAVs and Open-Source Photogrammetry Software for High- Resolution Monitoring of Alpine Glaciers: A Case Study from the Kanderfirn (Swiss Alps). Geosciences, 9, 356, 2019.
Hall, D. K. and Riggs, G. A.: MODIS/Terra Snow Cover Monthly L3 Global 0.05Deg CMG; Version 6; NASA National Snow and Ice Data Center Distributed Active Archive Center: Boulder, Co, USA, 2006.
Jóhannesson, T., Raymond, C. & Waddington, E. Time-scale for adjustment of glaciers to changes in mass balance. Journal of Glaciology 35, 355-369, 1989.
Kalnay, E., Kanamitsu, M., Kistler, R., Collins, W., Deaven, D., Gandin, L., Iredell, M., Saha, S., White, G.,Woollen, J., Zhu, Y., Leetmaa, A., Reynolds, R., Chelliah, M., Ebisuzaki,W., Higgins, W., Janowiak, J., Mo, K. C., Ropelewski, C.,Wang, J., Jenne, R., and Joseph, D.: The NCEP/NCAR 40-year reanalysis project, B. Am. Meteorol. Soc., 77, 437–470, https://doi.org/10.1175/1520-0477(1996)077<0437:TNYRP>2.0.CO;2, 1996.
Marzeion B, Cogley JG, Richter K, Parkes D. Glaciers. Attribution of global glacier mass loss to anthropogenic and natural causes. Science. 345(6199):919-21. doi: 10.1126/science.1254702, 2014
Whitehead, K., Moorman, B. J., and Hugenholtz, C. H.: Brief Communication: Low-cost, on-demand aerial photogrammetry for glaciological measurement, The Cryosphere, 7, 1879–1884, https://doi.org/10.5194/tc-7-1879-2013, 2013.

We thank the reviewer for the time dedicated to our manuscript and the detailed comments. We addressed these in our comments below (RC comments in red, AC in black).

Overview and general comments: I have read the manuscript entitled "Unprecedented loss of surface and cave ice in SE Europe related to record summer rains in 2019". In the manuscript, the authors investigate the ice changes in 5 subsurface glaciers (ice-caves) and 2 of the southernmost European glaciers during 2018-2019 year. Ice measurements in caves were performed using usual methods for ice body changes in ice caves by fixed marks/points in walls and photogrammetry, while glacier changes were evaluated using a drone. The weather during 2018-2019 period was evaluated using E-OBS dataset. The authors conclude that the observed ice loss during 2018- 2019 was caused by extreme rainfalls during spring-summer. This ice melt event was an unprecedented event, not recorded during the last century. The manuscript is interesting to understand the response of ice caves and glaciers to extreme rainfall events. Some minor observations and suggestions must be solved before publication.
Line 1: Title. The term "unprecedented" should be accompanied by a temporal term e.g ("Unprecedented loss of surface and cave ice in SE Europe since the last century. . ..or 100-years")

We thank the reviewer for the appreciation of our work and for the comments and suggestions. We welcome these and used them to improve our manuscript. General and point-by-point responses as well as modifications of the text are below (in black).

Following the reviewer's suggestion the title has been changed to better reflect the observations and conclusions: **Record summer rains in 2019 led to massive loss of surface and cave ice in SE Europe**

Lines 87 and 94: appear two times "Chionotrypa cave", I guess the caves have the same name, but maybe the authors could write directly the second name attributed "Chionotrypa (Falakro)" and "Chionotrypa (Olympus)" or, to shorten it, "Chionortypa-F" "Chionotrypa-O" to avoid repeating the name, as it appears along the text. An acronym for "Velika ledena jama v Paradani" is also recommended or for short like "Velika" or "Paradini"

In the first instance the names appear (lines 87 and 94) the text reads "*Chionotrypa Cave* (Mt. Falakro, hereafter *Chionotrypa Falakro*) and *Chionotrypa Cave* (Mt. Olympus, hereafter *Chionotrypa Olympus*)". We have removed the parentheses to make the names clearer and further checked the text to make sure that potential ambiguities are solved. We were thinking on adding acronyms, but in the end decided against (with several investigated sites, the text would have been to full of acronyms).

Line 104: change "metamorphosed" by "transformation"

Done

Line 126: The authors indicate that photogrammetry was carried out. In which caves was performed? Maybe the authors should add some additional information about the methods of photogrammetry, given that at the moment is not a usual method applied on ice caves, however, it is a powerful technique to know ice volume changes in caves.

We did so in Chionotrypa Falakro. We took photographs from the same point over successive years and the images were compared in image processing software. We added the following line to the main text:

"For the purpose of this paper, photographs of the upper surface of the ice and snow body in this cave taken from the same spot at the end of the ablation period were compared in order to visually estimate the ice level changes."

Line 136-142: A table could be helpful showing the resolution, error of MDT generation, and the accumulated error when comparing both models.

The following table was added to the main text

Table 1. Characteristics of the digital surface models and ortophotos obtained for Snezhnika and Banski Suhodol glacierets.

| Glacieret | Date of UAV flight | DSM resolution (cm) | Ortophoto Resolution (cm) | Mean error DSM (cm) | Mean error Ortophoto (cm) |
|---|---|---|---|---|---|
| **Snezhnika** | 28.10.2018 | 5 | 2.1 | 1.5 | 0.2 |
|  | 6.09.2019 | 22.5 | 2.8 | 2.1 | 0.3 |
| **Banski Suhodol** | 29.10.2018 | 10.6 | 4.9 | 1.9 | 0.3 |
|  | 5.09.2019 | 18.7 | 4.6 | 1.6 | 0.2 |

Line 144: I have asked an expert colleague in meteorology and he suggests some changes and modifications in relation to the use of the E-OBS dataset.

- One of the main problems of this dataset is related to the extreme events. The res- olution of E-OBS is 0.25x0.25, which for mountain areas and rainfall events is too low. The following database has been intensively tested: https://www.ecmwf.int/en/forecasts/datasets/reanalysis-datasets/era5

We agree with the concern of the reviewer, but E-OBS dataset actually has 2 different resolutions: 0.25 x 0.25 and 0.1 x 0.1. For the current analysis we have used the 0.1 x 0.1 resolution, which is the highest resolution you can obtain. We have opted for the E-OBS data set because is based on station data provided by each meteorological institute across Europe. Although ERA5 is also a very good and highly used dataset, for the precipitation it is more useful to use the E-OBS data set when we perform analysis over the European region. Moreover, The ERA5 precipitation production process does not include precipitation observation inputs, thus EOBS has a huge advantage over ERA5 in terms of precipitation. In general, the reanalysis datasets are preferred to observational dataset for regions where observational datasets have a limited coverage. But in the case of Europe, E-OBS offers the best alternative for precipitation.
EOBS data set link: https://surfobs.climate.copernicus.eu/dataaccess/access_eobs.php#datafiles

Line 167: Is it possible to indicate the drop or change in cm or m? Following a similar descriptions of the rest of ice caves where authors describe the drop of ice in cm or m, and then the corresponding volume.
We do not have precise measurements. This has been acknowledge in the revised manuscript:
In Chionotrypa Olympus, the surface of the ice is just 6 m below the cave entrance (Pennos et al., 2018) and thus the cave ice deposit responds to climatic variability in a manner similar to surface glaciers. The thermal inversion layer inside this shallow entrance shaft was easily destroyed during the prolonged warm spell, triggering the rapid melt of the surface and sides of the glacier; however, lack of precise measurements prevent us from estimating a figure for the volume of lost ice.

Line 198: Why did the authors use the 1971-2000 period instead of the last one (1980- 2010)?
The reference period is a matter of choice. We choose the period 1971 – 2000 because over the period 1981 – 2010 the global warming signal is much stronger, which might hinder the amplitude of the anomalies. Overall, most of climatological based studies use the period 1971 – 2000.

Line 307 "generate (semi)quantitative" add space.
Done

Figures
Figure 1: Add legend, Red stars: ice caves. Blue stars: glaciers
Done.

Figure 2: It would be nice to show all ice changes from the caves of the study in the graph maybe in a zoomed area.
We tried this, but the differences in resolution would have made the figure somehow awkward, with high resolution (monthly) in Scărișoara, seasonal in the other caves, and annual for the two surface glaciers.

Figure 4: The photos should be labelled as a). b) etc
Done.

Figure 5: The photos should be labelled with a code for a quick identification. Maybe left= 1 and corresponding photos 1a, 1b, 1c, Middle= 2a, 2b, 2c. . .etc. and similar for Figure 7.
Done.

Figure 8: indicate the meaning of background colors (warm, cold...).
Explanations for the color codes were added in the caption.

We thank the reviewer for the time dedicated to our manuscript and the detailed comments. We addressed these in our comments below (RC comments in red, AC in black).

Dear Authors, Dear Editor,
The manuscript reports enormous mass loss in five cave ice deposits and two small glaciers from SE Europe. Following the detailed evaluation of the meteorological conditions the Authors conclude that extreme precipitation events occurring in summer of 2019 led to catastrophic loss of ice. The study is well-structured and easy to follow; however, some revision can be recommended. I encourage the authors for the revision because the paper has the potential to became a key reference in the field of ice cave science.
best wishes, Zoltan Kern

Thank you for the detailed observations, comments and suggestions, we used them to improve the overall readability of the paper and clarify some of the potential ambiguities. Please find below our point-by-point responses and the resulting changing to the text.

General comments:
-I've found a bit confusing the usage of the term "glacier" in the paper. sometimes it seems that the Authors include both surface and cave glaciers in this term sometimes only surface glaciers. I suggest using the term "glacieret" when speaking about Snezhnika and Sudohol. It might help to avoid confusion.

Thank you for the suggestion, it is most welcome, as these ice bodies are indeed glacierets. This change would also help clarify some of the ambiguities in understanding our manuscript.

-I missed two highly relevant references from the discussion (Colucci et al., 2016, Colucci and Gugliemlin 2019), and suggested additional references or replacing the currently cited reference with a more pertinent one at a couple of places. Colucci RR, Fontana D, Forte E, et al. (2016) Response of ice caves to weather extremes in the Southeastern Alps, Europe. Geomorphology 261: 1–11. Colucci, R.R., Guglielmin, M. 2019 Climate change and rapid ice melt: Suggestions from abrupt permafrost degradation and ice melting in an alpine ice cave. Progress in Physical Geography 43: 561-573
https://doi.org/10.1177/0309133319846056

Thank your for reminding us of these two nice studies - we used them to better support our conclusions (see below).

-The name of one of the studied ice bodies is frequently written with a spelling mistake. "Basnki" should be corrected to "Banski" e.g. in line 114, 262, 266 or Fig4. I've marked this spelling mistake where I realized in the annotated PDF. I will not list them among the specific comments.

Corrected.

Although I marked few potential typos in the PDF, I note that I cannot provide a detailed linguistic review since I'm not a native English speaker.

Linguistic corrections were provided by a native speaker, dr. Sevasti Modestou (Canada/Norway).

specific comments:
line52: Citing this book chapter is not really relevant here, maybe the Authors wish to cite from the same year from the same author this paper: DOI:10.1017/RDC.2018.96

Done.

line62: A recent evaluation of CMIP5 simulations for this region (https://doi.org/10.1007/s00704-020-03361-7) confirmed the increase in winter precipitation, however showed that the models which reproduced better the decadal hydroclimate variability of the 1850-2005 period show less reduction in summer precipitation (instead increase). This suggests that it is more likely that summer precipitation will not decrease so highlight the importance of the topic of the study. It might be a relevant info here.

Thank you. We have used this and other references (Giorgi et al., 2011, 2016) to support these findings.

line68: I think Kern and Thomas 2014 is a more pertinent citation for this statement than Kern et al., 2013

Yes, it is also cited two lines above.

line76: Eisreisenwelt is not counted among the top5? If so then please, revise this statement.

For marketing reasons, Eisriesenwelt is considered to be "the largest ice cave in the world" – indeed it is the largest cave (42 km) *with* ice. The overall surface covered by ice is about 28,000 km$^2$ (Spoetl, 2018), but the volume is unknown and the ice does not occur as a single glacier, but as several distinct ice body. Anyway, it could be the 5$^{th}$ in terms of volume, after Dobsina (SK), Scarisoara, Focul Viu and Bortig (Ro), so we have modified the text accordingly.

line85: Is it really 'equilibrium' or just a moderately negative mass balance?

Compared with the tendency between 1947 and 1970s, we interpreted the post 1975 changes in ice level as indicating equilibrium, but the overall tendency is one of slight ice loss. We changed the relevant sentence as follows: "Monitoring of ice dynamics since 1947 (Persoiu and Pazdur, 2011) showed a rapid melt of ice during the 1950s due to changes in the morphology of the ice block, followed by alternating periods of ice growth and loss, superimposed on a moderate melting tendency."

line98: Geographical coordinates are lacking for Crna Ledenica.

Added now.

lines187-188: Are the described changes (retreat at the lower end of the two glacierets, increase in the width of the rimaye separating them from the cirque headwall) visible in Fig4? If yes, please refer to the figure.

Yes, they are. Ref to fig. 4 is being made in the text.

line238: Which zero isotherm do you mean? MAAT, Summer meant, July mean T? Please, be more specific.

Mean Annual Air Temperature. Info added in the text.

line238-241: I think a supporting reference for this statement is needed.

This is the case for all ice and snow accumulations at the bottom of shafts. With very few exceptions, ice in caves does not move horizontally, and when it does, it is only for several meters (at most) due to the restrictions imposed by the rock walls of the caves. We have inserted a reference to Perșoiu and Lauritzen, 2018 (Ice caves).

lines276-278: for instance, this is a place where Colucci et al., 2016, Colucci and Gugliemlin 2019 could be incorporated to the discussion.

Yes, indeed. We expanded the discussion to include these and other refernces.

Some additional minor comments are marked in the annotated PDF.

We have addressed all minor issues highlighted in the annotated PDF file.

Figures

Fig2: decimal places in the y-axis values can be omitted.

Done.

Fig3: The caption says that the changes are illustrated "since 2016", however the date of the first photo is 10.08.2014. Please, clarify this.

Typo, it is 2014

Fig6: I cannot see the panel codes in the figure. In addition, please, increase the characters in the title of the maps and use uniform character size for the titles.

Done.

Fig7: I cannot see the panel codes in the figure. In addition, please, increase the characters in the title of the maps and use uniform character size for the titles.

Done.

Fig8: Comment1: Could you add a color scale to help a better interpretation of the illustration. For instance, I guess reddish colors show warmer temp and bluish colors show colder temp, however it should be indicated in the caption. Comment2: Why the global temperature? European or SE European temperature changes could be much more relevant.

Explanations for the color codes were added in the caption. We used the global temperature, at it would make it easier for readers from different parts of the world to contextualize our findings.

Please also note the supplement to this comment: https://tc.copernicus.org/preprints/tc-2020-287/tc-2020-287-RC3-supplement.pdf

We have addressed all minor issues highlighted in the annotated PDF file.

---

## Referee Report (RR1)

**Review: "Record summer rains in 2019 led to massive loss of surface and cave ice in SE Europe"**

**Overview of the manuscript**

This manuscript investigates the changes of ice thickness of five ice-filled caves (Scărișoara, Chionotrypa-Falakro, Chionotrypa-Olympus, Crna Ledenica and Velika ledena jama v Paradani), as well as the area changes of two mountain glaciers (Snezhnika and Basnki Suhodol), all of them in Eastern Europe, during the hydrological year 2018-2019. The relatively large changes observed are associated to an anomaly in the weather (both summer and winter weather). The observations of ice changes are carried out based on in situ length measurements for the ice within caves, and with a drone for the two mountain glaciers. Weather parameters are obtained from the following datasets: E-OBS (Cornes et al., 2018), NCEP/NCAR (Kalnay et al., 1996) and MODIS/Terra Snow Cover Monthly L3 Global (Hall et al., 2006).

The authors have done substantial modifications of the manuscript, which has, in my opinion, improved significantly. They have also adequately addressed the majority of the questions from the first review. However, there are still a couple of aspects where the reply from the authors still raises some questions. This concerns mainly (1) the lack of error bars when presenting the results, (2) missing explanations in the methodology and (3) content of the abstract and conclusions in comparison to the actual content of the manuscript.

**General comments:**

**Error bars:**

Although the revised manuscript has improved in this regard, showing some kind of errors in the observations, still none of the results shows error bars. For the rigor of the study, this should be adequately addressed and there should be some sense of uncertainties in all the numbers, I.e. volumes and areas, should be provided in the results.

**Methodology:**

The revised manuscript shows a descriptive figure of the type of measurements done with tape. However, the authors do not explain how the measurements are carried out in Chionotrypa Falakro and Crna

Ledenica, where the Table 1 shows the measurements as "Photogrammetry". This needs a better explanation. Also, the drone survey data is now better presented, and a Digital Elevation Model (DEM) is shown for each survey, as well as a map of elevation difference. However, it should be straight-forward to present a volume change from their comparison (as well as the uncertainties associated), in the same way as volume changes are presented for the cave ice.

**Abstract and Conclusions:**

As pointed out in the first revision, there are some strong statements made in the abstract and conclusions that are not supported by the data and methods presented in this study, and they should not be presented in the abstract or in the conclusions, only as part of the discussion.

**Specific comments**

- L20-21: The role of changing atmospheric circulation patterns and distribution of precipitation in glacier changes is not studied as such in this manuscript. The atmospheric conditions are presented as observations but there is no strong link or model carried between these observations and the glacier changes. For this reason, I find this sentence outside the scope of the abstract.

- L29-30: Same as in the previous comment, this work does not analyze any climate model prediction, and therefore this should not be included in the abstract.

- L57: Throughout the manuscript the authors refer to "cave glacier", "ice cave" and "cave ice". This needs consistency.

- L149: Does the metal ruler allow sub-millimeter precision? (most rulers only allow measuring to one millimeter). This is contradicted in Table 1.

- L161-162: "Photographs of the upper surface of the ice body (…) to visually estimate the ice level changes". How exactly is this measure taken and how is it possible to obtain a precision of 0.3 cm with this method? See general comment nr2.

- L214: Mass balance vs volume change: The authors explained in their first author response that the dimension of "mass balance" yields a fundamental question and they decided to continue using this term at several occasions through the manuscript. I kindly disagree with this response, and in the revised manuscript I have not found a clear clarification of this, which still makes this

concept misleading. I suggest either adding a sentence to clarify what the authors define as mass balance, or replace expressions as "positive mass balance" for "volume gain".

- L226-230: Since the volume change is reported for the cave ice, why not reporting the volume change for the two glacierets too? This is straightforward to calculate from the map of elevation difference from the subtraction of the two DEMs.

- L343: Please provide a more updated reference for the accelerated warming of the Arctic.

- L353: This is still a very strong statement in the conclusions that is not supported by your material and methods. Therefore, it only should be mentioned in the discussion.

- L360-361: The Rossby waves have only been mentioned twice before in this manuscript and I also think this should not be a conclusion of your study. It needs to go to discussion.

**Figures**

- Fig. 5: As pointed out in the first round of comments, this figure showing changes in snow is heavily influenced by the seasonal differences of each picture. The authors argued that the pictures were taken at the end of the melting season, but this is not the case by looking at the date of acquisition. The snow field can change dramatically between July and October. This needs to be properly addressed or otherwise this figure should not be presented. Also, the "green" outline does not match with the "yellow" color in the legend. There seems to be a "green" line in the middle of the snow patch in the 2019 image.

- Fig. 7: For the soundness of the UAV work, it would be needed to show the comparison between the two DEMs in the entire overlapping area, and not just in the glacier itself. The color scale could also be improved: due to the high level of detail and accuracy in the DEMs, the scale bar could have more discrete steps in order to discern smaller changes in elevation.

**Tables**

- Fig. 5: Table 1: The uncertainties given here do not correspond to the values shown in the text. "Photogrammetry". Can you provide additional information on how the photogrammetric measurements are done in Chionotrypa Falakro and Crna Ledenica?

---

## Referee Report (RR2)

**Overview and general comments:**

I have read the revised version of the manuscript, previously entitled, *"Unprecedented loss of surface and cave ice in SE Europe related to record summer rains in 2019*", now entitled "*Record summer rains in 2019 led to massive loss of surface and cave ice in SE Europe*"

The new version of the manuscript has improved substantially with respect to the first one.

The authors provide a new figure about the morphology of the caves, which helps to have an idea about the ice location. In relation to the ice measurements in caves and glacierets, the authors include two new tables summarizing the place, the method used, the measurement uncertainty, the resolution, and the errors. This helps the reader to have a quick summary about the methods used to conduct the study. Moreover, the authors add a new digital elevation model of the glacierets studied showing the changes in ice surface elevation. They also provide a new fig. 3 to explain how the basal and superficial ice thaw is measured. Although these measurement methods are known in the ice caves community, this figure provides a visual example for non-expert readers. Does figure 3 show a real cave cross-section of the cave? if not, I suggest reducing the figure to a schematic illustration, something similar to the one attached below. The authors can use this or generate a new one if they want.

[Figure]

**Specific comments:**

Please see below some minor observations.

Line 186: To add space between the number and percentage (80% or 80 %) to be consistent.

In the figures, use the same letter source (e.g Fig 12 and Fig 11 have different letter sources). Also use parenthesis-letter- parenthesis (a), (b)… to numbering the figures. Some of them appear with/out parenthesis.

---

## Author Response (AR2)

**Responses to Reviewer 1**

Overview of the manuscript
This manuscript investigates the changes of ice thickness of five ice-filled caves (Scărișoara, Chionotrypa- Falakro, Chionotrypa-Olympus, Crna Ledenica and Velika ledena jama v Paradani), as well as the area changes of two mountain glaciers (Snezhnika and Basnki Suhodol), all of them in Eastern Europe, during the hydrological year 2018-2019. The relatively large changes observed are associated to an anomaly in the weather (both summer and winter weather). The observations of ice changes are carried out based on in situ length measurements for the ice within caves, and with a drone for the two mountain glaciers. Weather parameters are obtained from the following datasets: E-OBS (Cornes et al., 2018), NCEP/NCAR (Kalnay et al., 1996) and MODIS/Terra Snow Cover Monthly L3 Global (Hall et al., 2006).
The authors have done substantial modifications of the manuscript, which has, in my opinion, improved significantly. They have also adequately addressed the majority of the questions from the first review. However, there are still a couple of aspects where the reply from the authors still raises some questions. This concerns mainly (1) the lack of error bars when presenting the results, (2) missing explanations in the methodology and (3) content of the abstract and conclusions in comparison to the actual content of the manuscript.
Thank you for the detailed work on the manuscript and the appreciations. The responses and manuscript modifications are detailed below.

General comments:
Error bars:
Although the revised manuscript has improved in this regard, showing some kind of errors in the observations, still none of the results shows error bars. For the rigor of the study, this should be adequately addressed and there should be some sense of uncertainties in all the numbers, I.e. volumes and areas, should be provided in the results.
In the "Results" chapter, we added the values of the errors to the estimated volume changes for each cave glacier and surface glacieret.

Methodology:
The revised manuscript shows a descriptive figure of the type of measurements done with tape. However, the authors do not explain how the measurements are carried out in Chionotrypa Falakro and Crna Ledenica, where the Table 1 shows the measurements as "Photogrammetry". This needs a better explanation. Also, the drone survey data is now better presented, and a Digital Elevation Model (DEM) is shown for each survey, as well as a map of elevation difference. However, it should be straight-forward to present a volume change from their comparison (as well as the uncertainties associated), in the same way as volume changes are presented for the cave ice.
The following section of the "methods" describes the methodology (photo) for Chionotrypa Falakro and Crna Ledenica:
"In Chionotrypa Falakro, Chionotrypa Olympos and Crna Ledenica annual ice level fluctuations were intermittently recorded at the end of the ablation period over the past five years. Photographs of the upper surface of the ice and snow body taken intermittently from the same spot at the end of the ablation periods were compared in order to visually estimate the ice level changes. The errors associated with these measurements are less than 0.3 cm in Scărișoara Ice Cave and Velika ledena jama v Paradani, and are estimated to between 20 and 50 cm in Chionotrypa Falakro, Chionotrypa Olympos and Crna Ledenica (Table 1). The errors for the 2019 ice level changes are estimated between 1 and 25 % (Table 1), lower where rulers were used and higher where the presence of snow precluded precise measurements and/or estimates."

The volume changes for the two Bulgarian glacierets were calculated and the following line was added to the main text:
"The loss of ice at the surface of these glaciers (excluding thus basal melting) between 2018 and 2019 amounts to 10045 (±241) m3 (Banski Suhodol) and 2933 (129) m3 (Snezhnika)."

Abstract and Conclusions:
As pointed out in the first revision, there are some strong statements made in the abstract and conclusions that are not supported by the data and methods presented in this study, and they should not be presented in the abstract or in the conclusions, only as part of the discussion.
Specific comments
• L20-21: The role of changing atmospheric circulation patterns and distribution of precipitation in glacier changes is not studied as such in this manuscript. The atmospheric conditions are presented as observations but there is no strong link or model carried between these observations and the glacier changes. For this reason, I find this sentence outside the scope of the abstract.
We removed the reference to atmospheric circulation patterns and kept that on the distribution of precipitation, as this is what we have studied.
"While increasing air temperature is the main factor behind glacier mass and volume loss, variable patterns of precipitation distribution also play a role, though these are not as well understood."

• L29-30: Same as in the previous comment, this work does not analyze any climate model prediction, and therefore this should not be included in the abstract.

We did not analyze climate model predictions, as this was beyond the scope of the paper. We 1) presented and discussed the response of glaciers to extreme precipitation events and 2) present data (lines 373-385 of the revised ms) from previous studies that discusses model predications to contextualize our findings.

• L57: Throughout the manuscript the authors refer to "cave glacier", "ice cave" and "cave ice". This needs consistency.

Depending on context, we used these as follows:

Ice cave – refers to a rock cave hosting perennial ice deposits. We used "ice caves" when discussing the caves themselves. (e.g., "Ice caves occur in mountain regions across the Northern Hemisphere…").

Cave ice – refers to presence of ice in a cave, regardless of volume, mass, type etc (e.g. "Various proxies in cave ice have been used to reconstruct temperature variability")

Cave glacier – we used it when referring to the cave ice as affected by different processes (e..g., "average summer precipitation might play an important role in the overall volume changes of cave glaciers").

As these examples show, the three terms cannot be used interchangeably.

• L149: Does the metal ruler allow sub-millimeter precision? (most rulers only allow measuring to one millimeter). This is contradicted in Table 1.

Typo, thanks for spotting it! We mean "cm", not "mm".

• L161-162: "Photographs of the upper surface of the ice body (...) to visually estimate the ice level changes". How exactly is this measure taken and how is it possible to obtain a precision of 0.3 cm with this method? See general comment nr2.

The <0.3 cm precision refers to Scărișoara Ice Cave and Velika ledena jama v Paradani caves, only. For Chionotrypa Falakro, Chionotrypa Olympos and Crna Ledenica caves, the errors are 20 cm (Crna Ledenica), 30 cm (Chionotrypa Falakro) and 50 cm (Chionotrypa Olympos). These are the values given in table 1 and we have corrected the text for consistency (also deleted a duplicate part of the sentence). The full sentence reads:

The errors associated with these measurements are less than 0.3 cm in Scărișoara Ice Cave and Velika ledena jama v Paradani, and are estimated to between 20 and 50 cm in Chionotrypa Falakro, Chionotrypa Olympos and Crna Ledenica (Table 1).

• L214: Mass balance vs volume change: The authors explained in their first author response that the dimension of "mass balance" yields a fundamental question and they decided to continue using this term at several occasions through the manuscript. I kindly disagree with this response, and in the revised manuscript I have not found a clear clarification of this, which still makes this concept misleading. I suggest either adding a sentence to clarify what the authors define as mass balance, or replace expressions as "positive mass balance" for "volume gain".

While we used volume changes in the text, this was not done in a consistent manner throughout the manuscript. We corrected this and now the text reads volume gain/loss/change, as appropriate.

• L226-230: Since the volume change is reported for the cave ice, why not reporting the volume change for the two glacierets too? This is straightforward to calculate from the map of elevation difference from the subtraction of the two DEMs.

The volume changes for the two Bulgarian glacierets were calculated and the following line was added to the main text:

The loss of ice at the surface of these glaciers (excluding thus basal melting) between 2018 and 2019 amounts to 10045 m3 (Banski Suhodol) and 2933 m3 (Snezhnika).

• L343: Please provide a more updated reference for the accelerated warming of the Arctic.

Cohen et al., 2020

• L353: This is still a very strong statement in the conclusions that is not supported by your material and methods. Therefore, it only should be mentioned in the discussion.

We expanded the discussion with a new paragraph to support this inference.

• L360-361: The Rossby waves have only been mentioned twice before in this manuscript and I also think this should not be a conclusion of your study. It needs to go to discussion.

We stand behind this sentence, as it stems from our article.

Figures
• Fig. 5: As pointed out in the first round of comments, this figure showing changes in snow is heavily influenced by the seasonal differences of each picture. The authors argued that the pictures were taken at the end of the melting season, but this is not the case by looking at the date of acquisition. The snow field can change dramatically between July and October. This needs to be properly addressed or otherwise this figure should not be presented. Also, the "green" outline does not match with the "yellow" color in the legend. There seems to be a "green" line in the middle of the snow patch in the 2019 image.

The caves were (and are) regularly visited, during spring-autumn (risk of avalanches in winter prevents access to the surface of the ice). We take photos at each visit and use them to asses the dynamics of the upper face of the ice and snow body. The caves where these observations were made are located in Greece and S Croatia, where summers are dry, with only occasional rains (or, as was the case in 2019, prolonged periods of heavy thunderstorms). Thus, depending on meteo conditions, the ablation periods can be longer or shorter. For example, a very dry summer will shorten the ablation period, as 1) no warm rain water reaches the ice and 2) in the deep (~50 m) entrance shaft, the thermal inversion prevents the advection of heat to the surface of the ice while conduction through the air column results in limited warming. As a result (while present), ablation in dry summers can be extremely reduced, amounting to a few cm per several weeks, not discernable in visual/photogrammetric observations. We thus used those images that "signaled" the end of the ablation period for a specific year.

• Fig. 7: For the soundness of the UAV work, it would be needed to show the comparison between the two DEMs in the entire overlapping area, and not just in the glacier itself. The color scale could also be improved: due to the high level of detail and accuracy in the DEMs, the scale bar could have more discrete steps in order to discern smaller changes in elevation.
We made a new figure 7, according to the suggestions (comparison for the entire area covered by DEM, improved color scale)

Tables
• Fig. 5: Table 1: The uncertainties given here do not correspond to the values shown in the text. "Photogrammetry". Can you provide additional information on how the photogrammetric measurements are done in Chionotrypa Falakro and Crna Ledenica?
We corrected the errors in the main text.

**Responses to reviewer 2**

Overview and general comments:

I have read the revised version of the manuscript, previously entitled,

"Unprecedented loss of surface and cave ice in SE Europe related to record summer rains in 2019", now entitled "Record summer rains in 2019 led to massive loss of surface and cave ice in SE Europe"

The new version of the manuscript has improved substantially with respect to the first one.

The authors provide a new figure about the morphology of the caves, which helps to have an idea about the ice location. In relation to the ice measurements in caves and glacierets, the authors include two new tables summarizing the place, the method used, the measurement uncertainty, the resolution, and the errors. This helps the reader to have a quick summary about the methods used to conduct the study. Moreover, the authors add a new digital elevation model of the glacierets studied showing the changes in ice surface elevation. They also provide a new fig. 3 to explain how the basal and superficial ice thaw is measured. Although these measurement methods are known in the ice caves community, this figure provides a visual example for non-expert readers. Does figure 3 show a real cave cross-section of the cave? if not, I suggest reducing the figure to a schematic illustration, something similar to the one attached below. The authors can use this or generate a new one if they want.

Thank you for the comments and for the nice work behind the suggested figure, will use it.

Line 186: To add space between the number and percentage (80% or 80 %) to be consistent.

Done.

In the figures, use the same letter source (e.g Fig 12 and Fig 11 have different letter sources). Also use parenthesis-letter-parenthesis (a), (b)... to numbering the figures. Some of them appear with/out parenthesis.

All figures have been corrected where required to match the journal style.

**Responses to Reviewer 3**

Dear Authors, Dear Editor,
Thanks for the opportunity to read the revised version of the original manuscript. I appreciate the Authors' effort how they revised their original work. I think the introduction and the discussion improved a lot. The extension of the methodological section gives much better insight to the technical details of the applied methods which is a crucial requirement for the potential repeatability of the study. I like also the modified title of the study because it catches better the key message of the study. Since glacierets and cave ice deposits are currently the main representatives of the cryosphere in the studied region (SE Europe) I think the topic can be interesting to the readership of the journal.
As written above the study improved a lot compared to the original version, however some minor revisions are still needed before the publication of the study:

line 73: I think Munroe 2020 could be a more pertinent reference here than Kern and Persoiu 2013
Munroe, J. S.: First Investigation of Perennial Ice in Winter Wonderland Cave, Uinta Mountains, Utah, USA, The Cryosphere Discuss. [preprint], https://doi.org/10.5194/tc-2020-152, in review, 2020.
Added (it has since been published as Munroe (2021)).

line78: I suggest replacing "Most of" with "Many"
Changed to " Numerous caves hosting perennial ice"

lines 145,177: Please change "table" to "Table" in the brackets.
Done.

lines 307-308: Can you provide any evidence supporting the statement "The thermal inversion layer inside this shallow entrance shaft was easily destroyed during the prolonged warm spell,…"?
We noticed positive temperatures close to the surface of the ice during visits in summer. The text now reads: "The thermal inversion layer inside this shallow entrance shaft was easily destroyed during the prolonged warm spell (as observed during visits to the cave),"

lines 327-328: Needs revision. You probably wish to say that the climatic snowline (or TP-ELA) is above highest peaks in SE Europe, so these small glacierets are strongly controlled by topoclimatic conditions, and their mass balance is extremely affected by extreme events.
Yes, thank you for the clarification, we amended the text.

lines 331-332: Please consider replacing "glacierets we investigated" with "studied Bulgarian glacierets, ...." I think it is more understandable.
Indeed. Done.

line 353: "unraveled unprecedented" Sorry, the meaning is unclear to me.
We have investigated […] and [have] unraveled unprecedented

line 364: I suggest inserting "by the calibration-in-time approach" between the word "reconstructions" and "it"
We decided to keep sentence as it is now, its second part describes the 'calibration approach" (we hus avoid repetition)

line 366: I think "measured" should be written instead of "measure"
Indeed, thank you.

Figures
Fig2: This multipanel figure needs a careful checking. The cross section in panel 'e' seems to be Paradana however this cave is listed as 'd)' in the caption. Similarly, panel 'b' seems to be Chionotryopa Falakro, however it is listed as 'c)' in the caption. In addition, please check the name in the brackets after Paradana Cave. As far as I know the right spelling of the first surveyor is Pavel Kunaver.
Thank you for the sharp eye. We have corrected the references in the panels.
Fig7: In the caption "glaciarets" should be replaced with "glacierets"
Yes.
Fig8: Panel code should be 'e' instead of '3' in the caption.
Yes.
Fig12: in panel 'b' the histogram seems to be the same as in panel 'a' suggesting that the "no. of observations" were not tuned to the past 30years. Please check and correct it if needed.
We retained panel (a) only, as panel b does not bring new information.

---

## Author Response (AR3)

Editor Decision: Publish subject to technical corrections (07 Apr 2021) by Ketil Isaksen
Comments to the Author:
Dear authors,
We have received the revised manuscript and revision of your submission tc-2020-287 "Record summer rains in 2019 led to massive loss of surface and cave ice in SE Europe".
Given these revisions and responses and the improvements you made in the revised manuscript, I am pleased to accept your paper for publishing subject to technical corrections (see below) in The Cryosphere.
You will be hearing further from the journal concerning next steps in the publication process.
In the meantime, congratulations on the acceptance!
Thank you once again for considering The Cryosphere for the publication of your paper.

Sincerely,

Ketil Isaksen
Editor
The Cryosphere

Suggested changes:
P2L42 & P10L324-P11L339: Attention was also devoted to the role of short-term weather variability, extreme events and climate sensitivity in the study by Ødegård et al. (2017) to glacierets/ice patches. Consider to include the reference at appropriate places.

Ødegård, R. S., Nesje, A., Isaksen, K., Andreassen, L. M., Eiken, T., Schwikowski, M., and Uglietti, C.: Climate change threatens archaeologically significant ice patches: insights into their age, internal structure, mass balance and climate sensitivity, The Cryosphere, 11, 17–32, https://doi.org/10.5194/tc-11-17-2017, 2017.

P2L49: Suggest replacing "climate" with "microclimate"

Methods (P5-7): Suggests including subheadings for the different methods used

The manuscript needs some English smoothing.

Thank you for the editorial work. All changes suggested above were incorporated.